# Adipose tissue-secreted Spz5 promotes distal tumor progression via Toll-6-mediated Hh pathway activation in *Drosophila*

Du Kong [1,2,3,10] ✉, Xiaoqin Li[1,2,10], Sihua Zhao[1,2,10], Chenliang Wang[4,10], Zixin Cai [5], Sha Song[1,2], Yifan Guo [1,2], Xiaoyu Kuang[1,2], Xianping Wang [1,2], Wenhan Liu[1,2], Peng Liu[1,2], Xiaowei Guo[6], Wenyan Xu [1,2], Yirong Wang [5], Bin Zhao [4], Bin Jin [3,7,8]✉, Li He [9]✉ & Xianjue Ma [1,2]✉

## Abstract

**Interorgan communication is vital for tissue homeostasis and health in multicellular organisms, and its disruption can lead to diseases such as cancer. Adipose tissue acts as a key endocrine center, secreting cytokines that influence remote organs. Despite clear links between obesity and increased cancer risk, the underlying mechanisms are unclear. Here, utilizing a *Drosophila* genetic model combining Gal4-UAS and QF-QUAS tissue-specific transgene expression systems, we reveal that adipose-secreted Spz5 ligand promotes distal epithelial tumor overgrowth and invasion. Mechanistically, Spz5 binds to tumor cell Toll-6 receptors, triggering the degradation of the endocytic adaptor protein AP-2α via Mib1-mediated ubiquitination. Consequently, impaired endocytosis leads to Smoothened (Smo) accumulation on the cell membrane and subsequent activation of the Hedgehog (Hh) pathway. This abnormal Hh activation synergizes with the oncogenic Yorkie (Yki) to drive tumor growth and invasion. Furthermore, tumor-derived Unpaired ligands (Upds) activate the JAK-STAT pathway in the fat bodies, which leads to Hippo pathway-dependent upregulation of *spz5* expression. Thus, our study provides insights into the complex regulatory mechanisms by which systemic interorgan communication influences tumor progression.**

Keywords Toll-6; Fat Body; Interorgan; Hedgehog; *Drosophila*
Subject Categories Cancer; Signal Transduction

## Introduction

Interorgan communication is crucial for both normal development and maintenance of homeostasis in multicellular organisms. Disruption of this communication can lead to various diseases, including cancer. Tumor cells possess the ability to secrete various cytokines that can affect the functions of distant organs, while other tissues and organs can also influence the occurrence and progression of distant tumors, a phenomenon known as tumor-host interactions (Bilder et al, 2021). Adipose tissue, in addition to its role in energy storage, functions as an important endocrine organ, releasing a broad range of cytokines such as FGF2, TNF-α, IL-1β, and IL-6 that are closely associated with human liver, stomach, colon, pancreas, or breast cancers (Aune et al, 2012; Cascetta et al, 2018; Chakraborty et al, 2017; Dirat et al, 2011; Murphy et al, 2018). Despite the substantial epidemiological evidence linking the accumulation of adipose tissue to increased risks of multiple cancers, the complex effects and underlying mechanisms of adipose-derived factors on tumors remain elusive.

*Drosophila melanogaster* possesses organ systems that share remarkable similarities with those of humans. These organs not only have unique functions but are also interconnected, collectively maintaining organismal homeostasis (Ugur et al, 2016). Furthermore, the genes and signaling pathways associated with human diseases, particularly cancer, are highly conserved between fruit flies and humans, making *Drosophila* an excellent model for studying interorgan communication between tumors and other tissues (Bilder et al, 2021; Ding et al, 2021; Galikova and Klepsatel, 2018; Ghosh et al, 2020; Kwon et al, 2015; Liu et al, 2022b; Newton et al, 2020; Pastor-Pareja and Xu, 2011; Singh et al, 2024; Song et al, 2017a; Song et al, 2019; Song et al, 2017b). The *Drosophila* fat body, composed of large polyploid cells (adipocytes), plays a crucial role in lipid storage, energy metabolism, innate immunity, and

[1]School of Life Sciences, Westlake University, 310024 Hangzhou, Zhejiang, China. [2]Westlake Laboratory of Life Sciences and Biomedicine, 310024 Hangzhou, Zhejiang, China. [3]Department of Hepatobiliary Surgery, The Second Hospital, Cheeloo College of Medicine, Shandong University, 250033 Jinan, Shandong, China. [4]MOE Key Laboratory of Biosystems Homeostasis & Protection, Zhejiang Provincial Key Laboratory for Cancer Molecular Cell Biology, and Innovation Center for Cell Signaling Network, Life Sciences Institute, Zhejiang University, 310058 Hangzhou, Zhejiang, China. [5]College of Biology, Hunan University, 410082 Changsha, Hunan, China. [6]The Key Laboratory of Model Animals and Stem Cell Biology in Hunan Province, School of Medicine, Hunan Normal University, 410013 Changsha, Hunan, China. [7]Organ Transplant Department, Qilu Hospital of Shandong University, 250033 Jinan, Shandong, China. [8]Shandong Province Engineering Research Center for Multidisciplinary Research on Hepatobiliary and Pancreatic Malignant Tumors, 250033 Jinan, Shandong, China. [9]The First Affiliated Hospital of USTC, Division of Life Sciences and Medicine, University of Science and Technology of China, 230000 Hefei, Anhui, China. [10]These authors contributed equally: Du Kong, Xiaoqin Li, Sihua Zhao, Chenliang Wang. ✉E-mail: kongdu@email.sdu.edu.cn; jinbin@sdu.edu.cn; lihe19@ustc.edu.cn; maxianjue@westlake.edu.cn

detoxification (Arrese and Soulages, 2010; Buchon et al, 2014; Gade, 2004; Li et al, 2019; Li et al, 2007). Recent studies have unveiled the fat body's ability to sense various hormones and nutrient signals, regulating larval growth, body size, circadian rhythms, pupal diapause, lifespan, and feeding behavior by releasing regulatory molecules called fat body signals (FBSs) or adipokines (Arrese and Soulages, 2010; Lemaitre and Hoffmann, 2007; Li et al, 2019; Zheng et al, 2016). Emerging evidence suggests that the fat body can remotely regulate tumor progression in *Drosophila* larval epithelial tissues (de Vreede et al, 2022; Kong et al, 2022; Parisi et al, 2014). To further elucidate the crosstalk mechanism between the fat body and epithelial tumors, we established an in vivo model to examine this communication by integrating the *Drosophila* FLP-FRT, Gal4-UAS, and QF-QUAS systems. Specifically, we induced malignant mosaic tumors on larval imaginal discs using the QF-QUAS-based QMARCM technique. Simultaneously, we manipulated gene expression in the fat body using the Gal4-UAS system. Through this model, we discovered that the fat body-secreted ligand, Spz5, acts through the Toll-6 receptor on tumor surfaces, resulting in abnormal activation of the Hedgehog pathway and subsequent tumor progression. In turn, the ligands derived from tumors, Upd1/2/3, activate the JAK-STAT signaling pathway in the fat body, promoting the secretion of Spz5 from the fat body in a Hippo pathway-dependent manner. This positive feedback loop between the epithelial tumor and the fat body, mediated by Upds and Spz5, collectively facilitates tumor progression.

The Hedgehog (Hh) pathway is highly conserved between fruit flies and humans and plays a critical role in embryonic development and adult tissue homeostasis (Ingham and McMahon, 2001; Ingham et al, 2011; Jiang and Hui, 2008; Nusslein-Volhard and Wieschaus, 1980). Dysregulation of Hh signaling is associated with numerous human diseases, including congenital defects and cancer (Jiang, 2022; Pasca di Magliano and Hebrok, 2003; Taipale and Beachy, 2001). The core components of the Hh signaling pathway in *Drosophila* comprise the Hh receptor Patched (Ptc), the signal transducer Smoothened (Smo), and the zinc finger transcription factor Cubitus interruptus (Ci) (Camp et al, 2010; Zheng et al, 2010). Upon binding of Hh to its receptor Ptc, the inhibition of Smo by Ptc is relieved, leading to the accumulation and activation of Smo on the cell surface. This, in turn, activates the transcription factor Ci to regulate the expression of Hh target genes (Jiang and Hui, 2008; Pasca di Magliano and Hebrok, 2003). Smo is a G-protein-coupled receptor (GPCR) essential for transducing Hh signals across the plasma membrane in both insects and vertebrates (Ingham and McMahon, 2001; Zhao et al, 2007). The regulation of Smo localization and activity is closely associated with the endocytosis process. In the absence of the Hh ligand, Smo is internalized from the plasma membrane through clathrin-coated pits and directed to degradation in the lysosomes. This mechanism keeps the Hh pathway inactive (Jia and Jiang, 2022). Disruptions in endocytosis can cause abnormal accumulation and activation of Smo, leading to aberrant Hh signaling and the development of various cancers, such as basal cell carcinoma (BCC) and medulloblastoma (Jiang, 2022). As a result, Smo represents an attractive therapeutic target. In this study, we unveiled that activated Toll-6, by binding to the E3 ligase Mib1, triggers AP-2α ubiquitination and degradation, impairing endocytosis. This disruption causes Smo accumulation, activating the Hedgehog

pathway, which collaborates with oncogenic Yorkie (Yki) to fuel tumor growth and invasion. Our findings, supported by an in vivo interorgan communication model, elucidate adipose tissue's role in epithelial tumor progression and deepen understanding of regulatory mechanisms in tumorigenesis.

# Results

## Adipose-derived Spz5 remotely promotes Toll-6-dependent epithelial tumor progression

Loss of tumor suppressor genes and activation of oncogenes contribute significantly to tumor development (Igaki et al, 2006; Pagliarini and Xu, 2003). In *Drosophila*, homozygous mutant discs for the cell polarity gene *scribble* (*scrib*) develop into large tumorous masses (Bilder et al, 2000; Pagliarini and Xu, 2003; Yamamoto et al, 2017). However, *scrib* mutant clones are actively eliminated through tumor-suppressive cell competition and exhibit reduced size compared to the wild-type (*WT*) clones (Fig. 1A,B) (Chen et al, 2012; Katsukawa et al, 2018; Ohsawa et al, 2011; Yamamoto et al, 2017). Conversely, *yorkie* (*yki*), the transcription cofactor of the Hippo pathway, acts as an oncogene and is highly expressed in various human cancers (Kong et al, 2021; Pagliarini and Xu, 2003). Consistent with previous reports, overexpression of the wild-type *yki* (*Qyki^{WT}*) using the QMARCM system is sufficient to rescue *scrib^{-/-}* clones from being outcompeted without forming a malignant tumor (Fig. 1A–C) (Chen et al, 2012). Furthermore, although overexpressing the constitutively activated form of *yki* (*Qyki^{ACT}*) alone induces dramatic clonal overgrowth, it does not exhibit high malignancy, as demonstrated by the limited increase in the invasion rate of ventral nerve cord (VNC), an indicator of malignancy in fly tumors (Fig. 1A–C) (Pagliarini and Xu, 2003). Interestingly, a strong synergetic tumor-promoting effect was observed between *scrib^{-/-}* and *Qyki^{ACT}*, resulting in a significant increase in tumor overgrowth and VNC invasion (Fig. 1A–C). In addition, the expression of Matrix metalloproteinase (Mmp1), a well-established marker for tumor invasion (Pagliarini and Xu, 2003), was significantly upregulated in the invasive leading edges of *Qyki^{ACT}/scrib^{-/-}* clones compared to *WT* or *Qyki^{ACT}* clones (Fig. EV1A).

We have previously identified the tumor-suppressive function of the Spz5-Toll-6 axis and found that Spz5, secreted by the fat body, binds to Toll-6 to facilitate the elimination of premalignant *scrib^{-/-}* cells (Kong et al, 2022). Interestingly, our antibody staining assay against Toll-6 and also revealed a significant upregulation of Toll-6 protein levels in *Qyki^{ACT}/scrib^{-/-}* tumors compared to *Qyki^{ACT}* clones alone (Fig. 1D,E), suggesting a potential functional role of Spz5-Toll-6 axis in *Qyki^{ACT}/scrib^{-/-}* malignant tumors. Surprisingly, depleting *Toll-6* within *Qyki^{ACT}/scrib^{-/-}* malignant tumors had an unexpected effect: it inhibited tumor growth and invasion (Figs. 1F–H and EV1B). This contradicts the observation of *Toll-6*-depleted *scrib^{-/-}* clone promoting growth in the earlier cell competition scenario (Kong et al, 2022).

To understand the contradictory role played by the Spz5-Toll-6 axis in the *Qyki^{ACT}/scrib^{-/-}*-induced malignancy condition, it is essential to establish a novel genetic model that enables *Qyki^{ACT}/scrib^{-/-}* malignant tumor induction in the E-A disc. In addition, this model must support simultaneous genetic manipulation

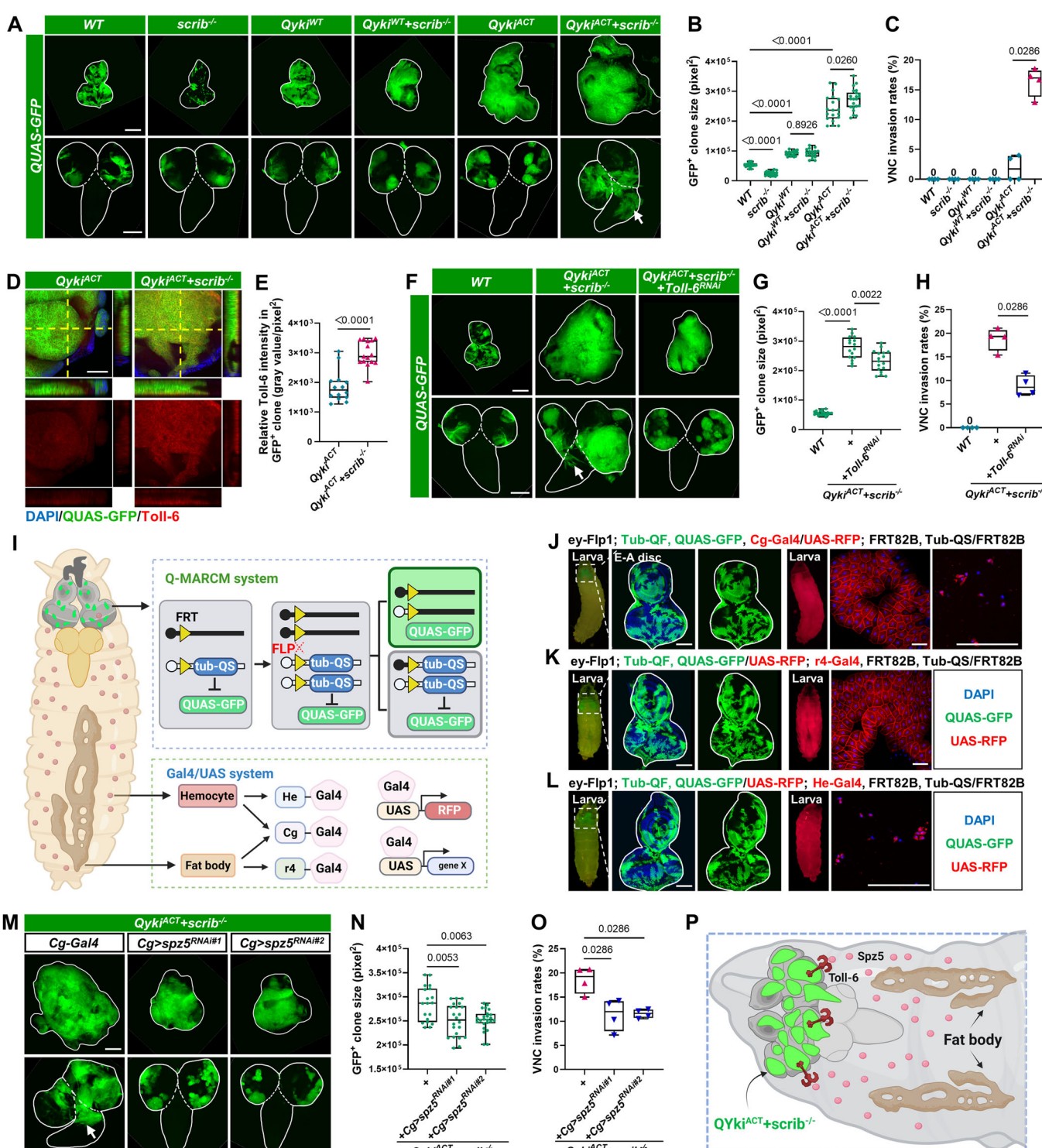

**Figure**

through the use of an alternate binary expression system. *Drosophila* is equipped with powerful genetic tools for gene expression, including the Gal4-UAS, QF-QUAS, and LexA-lexAop systems. The Gal4-UAS system, in combination with the FLP-FRT system and the inhibitory component Gal80, is commonly used to construct the Mosaic Analysis with a Repressible Cell Marker (MARCM) technique for mosaic clonal analysis (Lee

and Luo, 1999; Wu and Luo, 2006; Xu and Rubin, 1993; Zirin et al, 2024). Similarly, the QMARCM system can be established by combining the QF-QUAS system with its inhibitory elements QS and the FLP-FRT (Potter et al, 2010). Recently, several studies have utilized these different systems to investigate interorgan communication in *Drosophila* (Hodgson et al, 2021; Lodge et al, 2021; Xu et al, 2023). Therefore, we established a genetic model by

◄

**Figure 1. Adipose-derived Spz5 promotes epithelial malignant tumor progression.**

(A) Eye-antennal discs (E-A discs, top) and ventral nerve cord (VNC, bottom) with GFP-labeled mosaic clones induced by ey-FLP/QMARCM in wild-type *(WT)*, *scrib*$^{-/-}$, *Qyki*$^{WT}$ *(QUAS-yki*$^{WT}$*)*, *Qyki*$^{WT}$*/scrib*$^{-/-}$, *Qyki*$^{ACT}$ *(QUAS-yki*$^{ACT}$*)*, and *Qyki*$^{ACT}$*/scrib*$^{-/-}$ genotypes. The white lines outline the borders of the E-A discs (top) and the VNC (bottom). The arrow indicates VNC invasion site. (B, C) Quantification of GFP-labeled clone size in E-A discs (B, from left to right, $n = 23, 22, 19, 18, 19, 22$), or percentage of clone invasion into VNC (C, total number from four independent experiments, from left to right, $n = 82, 106, 89, 91, 106, 117$) for the indicated genotypes. (D) z-stack confocal images of E-A discs with GFP-labeled *Qyki*$^{ACT}$ (left) or *Qyki*$^{ACT}$*/scrib*$^{-/-}$ (right) clones were stained with anti-Toll-6 antibody. (E) Quantification of the relative intensity of Toll-6 in clones with the indicated genotypes (from left to right, $n = 14, 15$). (F) E-A discs (top) and VNC (bottom) with GFP-labeled clones induced by ey-FLP/QMARCM in *WT*, *Qyki*$^{ACT}$*/scrib*$^{-/-}$, and *Qyki*$^{ACT}$*/scrib*$^{-/-}$*/Toll-6*$^{RNAi}$ genotypes. The white lines outline the borders of the E-A discs (top) and the VNC (bottom). The arrow indicates VNC invasion site. (G, H) Quantification of GFP-labeled clone size in E-A discs (G, from left to right, $n = 20, 16, 17$), or percentage of clone invasion into VNC (H, total number from four independent experiments, from left to right, $n = 110, 102, 113$) for the indicated genotypes. (I) Cartoon illustrating the strategy for studying the interorgan communication between fat body or hemocytes and E-A discs using the *Drosophila* QMARCM and Gal4/UAS systems. The QMARCM system generates random clones expressing QUAS-controlled genes in the larval E-A discs, whereas the Gal4/UAS system expresses UAS-controlled genes in either fat body or hemocytes. (J–L) E-A discs with mosaic clones induced by ey-FLP/QMARCM technique, marked by the presence of GFP. UAS-RFP is expressed in both fat body and hemocytes by *Cg-Gal4* (J), in fat body by *r4-Gal4* (K), and in hemocytes by *He-Gal4* (L). The white lines outline the borders of the E-A discs. (M) E-A discs (top) and VNC (bottom) with GFP-labeled *Qyki*$^{ACT}$*/scrib*$^{-/-}$ clones, and expression of *WT*, *spz5*$^{RNAi#1}$, and *spz5*$^{RNAi#2}$ in fat body and hemocytes under control of *Cg-Gal4*. The white lines outline the borders of the E-A discs (top) and the VNC (bottom). The arrow indicates VNC invasion site. (N, O) Quantification of GFP-labeled clone size in E-A discs (N, from left to right, $n = 19, 22, 24$), or percentage of clone invasion into VNC (O, total number from four independent experiments, from left to right, $n = 111, 100, 94$) for the indicated genotypes. (P) A schematic representation of the crosstalk between fat body and E-A discs bearing GFP-labeled *Qyki*$^{ACT}$*/scrib*$^{-/-}$ clones. The $P$ values of (B, C, E, G, H, N, O) were determined by unpaired nonparametric Mann–Whitney test. Exact $P$ values are shown in the figures. The box plots of (B, C, E, G, H, N, O) boundaries represent the 25th (lower quartile) and 75th (upper quartile) percentiles, with the center line indicating the median, and the whiskers extend to the minimum and maximum values. Scale bars: 200 μm for (A, F, M); 100 μm for (J, K, L); 50 μm for (D). DAPI 4′,6-diamidino-2-phenylindole. Source data are available online for this figure.

combining the QMARCM system and Gal4-UAS system to dissect interorgan communications between non-tumor tissues and epithelial tumors. Specifically, we generate GFP-labeled QF-expressing mosaic clones on the eye-antennal imaginal disc (E-A disc) using *ey-Flp* activated QMARCM system, while manipulating gene expression in other tissues by using the Gal4-UAS system, which is commonly employed (Fig. 1I). We validated the efficacy and specificity of this system across different cell types, including adipose tissue and hemocytes. We observed GFP$^+$ mosaic clones in the E-A disc and simultaneously detected specific RFP expressions in the fat body and/or hemocytes (Fig. 1J–L).

Next, we utilized this QMARCM-induced *Qyki*$^{ACT}$*/scrib*$^{-/-}$ tumor model to dissect the in vivo role of Spz5 in distal organs (Fig. EV1C). In line with the potential tumor-promoting role of the Spz5-Toll-6 axis, we found that knockdown of *spz5* in adipose tissue using the *Cg-Gal4* (labels both adipose tissue and hemocytes) or the adipose tissue-specific promoter *r4-Gal4* significantly inhibited the overgrowth and VNC invasion of *Qyki*$^{ACT}$*/scrib*$^{-/-}$ tumors (Figs. 1M–O and EV1D–F). In contrast, *spz5* knockdown using the hemocyte-specific *He-Gal4* did not affect the growth and invasion of *Qyki*$^{ACT}$*/scrib*$^{-/-}$ tumors (Fig. EV1G–I). These results indicate that Spz5 in adipose tissue, rather than in hemocytes, promotes the growth and invasion of *Qyki*$^{ACT}$*/scrib*$^{-/-}$ tumors (Fig. 1P). Considering our previous findings that adipose tissue facilitates the clearance of premalignant *scrib*$^{-/-}$ clones through the Spz5-Toll-6 axis, the above data suggest that hyperactivated Yki may hijack this tumor-suppressive role of Toll-6 and instead promote tumor malignancy.

## Activated Toll-6 collaborates with oncogenic Yki to promote tumor progression

Next, we investigated whether Toll-6 activation is sufficient to collaborate with hyperactivated Yki to drive tumor malignancy. In line with Toll-6's tumor-promoting function in *Qyki*$^{ACT}$*/scrib*$^{-/-}$ tumors, clonal overexpression of an activated form of *Toll-6* (*Toll-6*$^{ACT}$) in the E-A disc was sufficient to synergize with *Qyki*$^{ACT}$, *yki*$^{S168A}$, or *yki*$^{S111A.S168A.S250A}$ (*yki*$^{3SA}$) to induce malignancy, as

characterized by increased tumor volume, enhanced invasive migration toward VNC, decreased pupation rate, and formation of Mmp1$^+$ protrusions toward the VNC (Figs. 2A–E and EV2A–F). In contrast, clonal expression of *Qyki*$^{ACT}$, *yki*$^{S168A}$, or *yki*$^{3SA}$ alone in the E-A disc had a slight effect on VNC invasion when compared with controls (Figs. 2A–C and EV2A–D). Moreover, we also observed a similar synergistic effect between *Toll-6*$^{ACT}$ and *yki*$^{ACT}$ in the developing wing disc (Fig. EV2G–L), indicating a universal role of *yki*$^{ACT}$*/Toll-6*$^{ACT}$ in driving epithelium tumor malignancy.

Sustained proliferation, apoptosis evasion, and disrupted differentiation are key hallmarks of cancer (Hanahan, 2022). In line with this, we observed a significant increase in PH3, the proliferation marker, in *yki*$^{S168A}$*/Toll-6*$^{ACT}$ tumors, compared with controls (Figs. 2F and EV2M). We also found that *Toll-6*$^{ACT}$-induced apoptosis was notably suppressed by co-expressing activated *yki* (Figs. 2G and EV2N). In addition, overexpression of *Toll-6*$^{ACT}$ dramatically decreased the expression of differentiation markers, including Elav and Hindsight (Fig. EV2O–Q) (Pickup et al, 2009). Collectively, these data suggest that in the absence of activated *yki*, activated *Toll-6* promotes *scrib*$^{-/-}$ cell clearance through apoptosis (Kong et al, 2022). However, in the presence of activated *yki*, it counteracts *Toll-6*-induced apoptosis and synergistically promotes malignant tumor progression (Fig. 2H).

## Activated Toll-6 promotes Yki-driven tumorigenesis by activating Hh pathway

Next, we explored the underlying mechanisms by which activated *Toll-6* induces malignancy in *yki*$^{ACT}$ tumors. We have previously conducted a small-scale antibody screen to examine the activation of several growth-regulating pathways within the *Toll-6*$^{ACT}$ clones, including JNK, Hippo, and JAK-STAT pathways (Kong et al, 2022). Here, we also examined additional growth-regulating pathways, including Notch, EGFR, and Hedgehog. We previously demonstrated that JAK-STAT pathway activation was not significantly affected by the overexpression of *Toll-6*$^{ACT}$, whereas JNK signaling was moderately activated (Kong et al, 2022). Furthermore, ectopic expression of *Toll-6*$^{ACT}$ robustly downregulated multiple Yki targets,

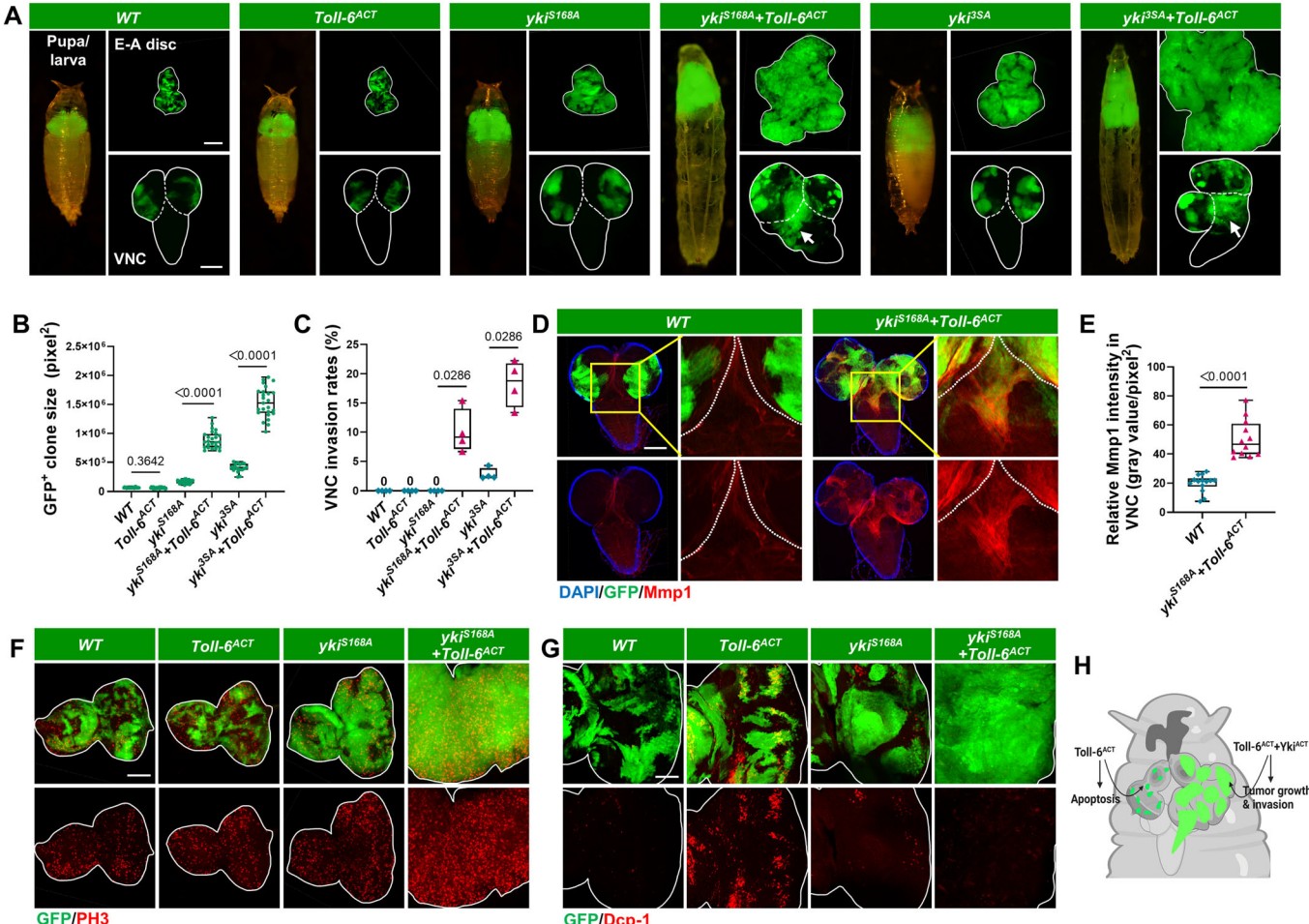

**Figure 2. Activated Toll-6 collaborates with oncogenic Yki to promote tumorigenesis.**

(A) Pupa/larva (left), E-A discs (right, top), and VNC (right, bottom) with GFP-labeled clones induced by ey-FLP/MARCM in *WT*, *Toll-6^ACT^*, *yki^S168A^*, *yki^S168A^/Toll-6^ACT^*, *yki^3SA^*, and *yki^3SA^/Toll-6^ACT^* genotypes. The white lines outline the borders of the E-A discs (right, top) and the VNC (right, bottom). The arrows indicate VNC invasion sites. (B, C) Quantification of GFP-labeled clone size in E-A discs (B, from left to right, $n = 24, 28, 25, 25, 18, 25$), or percentage of clone invasion into VNC (C, total number from four independent experiments, from left to right, $n = 61, 98, 129, 172, 177, 175$) for the indicated genotypes. (D) VNC with GFP-labeled clones induced by ey-FLP/MARCM for *WT* and *yki^S168A^/Toll-6^ACT^* were stained with anti-Mmp1 antibody. White dashed lines in the magnified image mark the boundary between the optic lobe and the VNC. (E) Quantification of the relative Mmp1 intensity in VNC with indicated genotypes (from left to right, $n = 14, 12$). (F) E-A discs with GFP-labeled clones induced by ey-FLP/MARCM for *WT*, *Toll-6^ACT^*, *yki^S168A^*, and *yki^S168A^/Toll-6^ACT^* were stained with anti-phospho-histone H3 (PH3) antibody. The white lines outline the borders of the E-A discs. (G) E-A discs with GFP-labeled clones induced by ey-FLP/MARCM for *WT*, *Toll-6^ACT^*, *yki^S168A^*, and *yki^S168A^/Toll-6^ACT^* were stained with anti-cleaved *Drosophila* Dcp-1 antibody. The white lines outline the borders of the E-A discs. (H) A diagram illustrates that activated Toll-6 exhibits completely opposite effects—pro-apoptotic and pro-proliferative—under different conditions. The P values of (B, C, E) were determined by unpaired nonparametric Mann–Whitney test. Exact P values are shown in the figures. The box plots of (B, C, E) boundaries represent the 25th (lower quartile) and 75th (upper quartile) percentiles, with the center line indicating the median, and the whiskers extend to the minimum and maximum values. Scale bars: 200 μm for (A, D); 100 μm for (F); 50 μm for (G). DAPI 4′,6-diamidino-2-phenylindole. Source data are available online for this figure.

including Diap1 (Kong et al, 2022). Interestingly, we detected a significant increase in punctate-like accumulation of NICD (Notch Intracellular Domain) and NECD (Notch Extracellular Domain) within *Toll-6^ACT^* clones (Fig. EV3A,B). However, no significant changes were observed in the expression of the Notch target gene Cut or its reporter NRE-GFP in *Toll-6^ACT^* clones (Fig. EV3C,D). We also examined EGFR activation using a pERK-specific antibody, as well as the expression of EGFR target genes, including *spi* and *pnt*. We observed a significant downregulation of phosphorylated ERK in *Toll-6^ACT^* clones (Fig. EV3E), while the target genes of EGFR remained unaffected (Fig. EV3F–H). Conversely, *Toll-6^ACT^* clones

showed a marked upregulation of the Hh pathway's key transcription factor Ci (Fig. 3A), and this upregulation was more pronounced in *yki^ACT^/Toll-6^ACT^* tumors than in *yki^ACT^* alone (Fig. EV3I). Accordingly, several target genes of Hh pathway including *Ptc* and *dpp* were upregulated in Toll-6-activated clones (Fig. 3B,C), indicating the activation of the Hh pathway upon *Toll-6^ACT^* expression.

Given that the Hh pathway has been found to promote apoptosis and tumorigenesis (Jiang, 2022; Liu et al, 2021), similar to Toll-6 activation, we aimed to examine the role of Hh signaling in Toll-6-mediated tumor progression. In line with this, effective

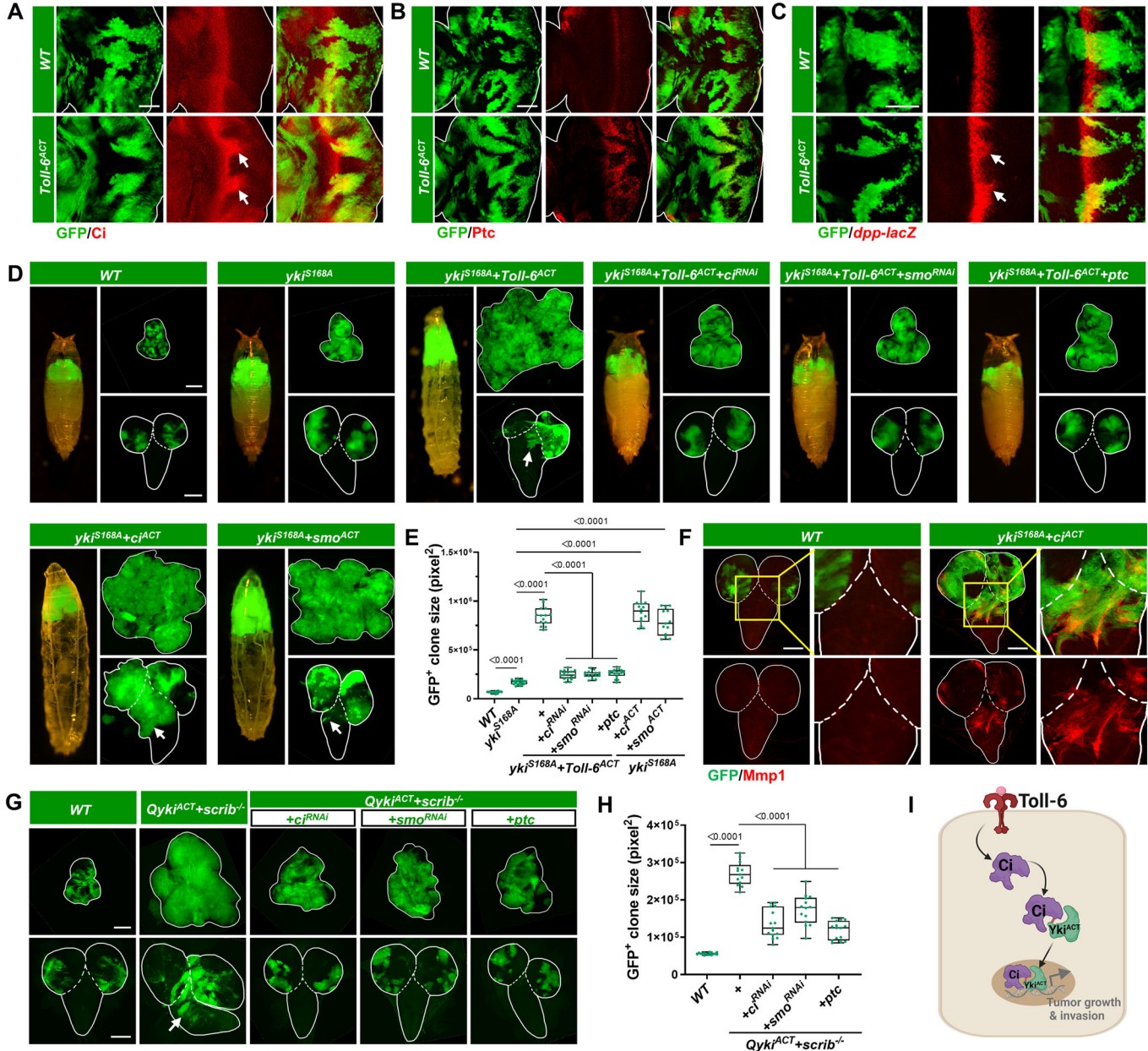

**Figure 3. Activated Toll-6 promotes tumor malignancy by activating the Hh signaling pathway.**

(A–C) E-A discs with GFP-labeled *WT* (top) and *Toll-6^ACT* (bottom) clones were stained with anti-Ci antibody (A), anti-Ptc antibody (B), and anti-β-galactosidase to label *dpp* transcription (C). The white lines in (A, B) outline the borders of the E-A discs. The arrows in (A) indicate increased Ci protein level, and the arrows in (C) indicate increased *dpp* transcription level. (D) Pupa/larva (left), E-A discs (right, top), and VNC (right, bottom) with GFP-labeled mosaic clones induced by ey-FLP/MARCM in *WT*, *yki^S168A*, *yki^S168A/Toll-6^ACT*, *yki^S168A/Toll-6^ACT/ci^RNAi*, *yki^S168A/Toll-6^ACT/smo^RNAi*, *yki^S168A/Toll-6^ACT/ptc*, *yki^S168A/ci^ACT*, and *yki^S168A/smo^ACT* genotypes. The white lines outline the borders of the E-A discs (right, top) and the VNC (right, bottom). The arrows indicate VNC invasion sites. (E) Quantification of GFP-labeled clone size in E-A discs with indicated genotypes (from left to right, n = 15, 18, 14, 16, 14, 16, 14, 13). (F) VNC with GFP-labeled clones induced by ey-FLP/MARCM for *WT* and *yki^S168A/ci^ACT* were stained with anti-Mmp1 antibody. White dashed lines in the magnified image mark the boundary between the optic lobe and the VNC. (G) E-A discs (top) and VNC (bottom) with GFP-labeled clones induced by ey-FLP/QMARCM for *WT*, *Qyki^ACT/scrib^−/−*, *Qyki^ACT/scrib^−/−/ci^RNAi*, *Qyki^ACT/scrib^−/−/smo^RNAi*, and *Qyki^ACT/scrib^−/−/ptc*. The white lines outline the borders of the E-A discs (top) and the VNC (bottom). The arrow indicates VNC invasion site. (H) Quantification of GFP-labeled clone size in E-A discs with indicated genotypes (from left to right, n = 16, 13, 15, 14, 15). (I) A schematic representation of activated Toll-6 activates Hedgehog signaling to promote Yki-driven tumorigenesis. The *P* values of (E, H) were determined by unpaired nonparametric Mann–Whitney test. Exact *P* values are shown in the figures. The box plots of (E, H) boundaries represent the 25th (lower quartile) and 75th (upper quartile) percentiles, with the center line indicating the median, and the whiskers extend to the minimum and maximum values. Scale bars: 200 μm for (D, F, G); 50 μm for (A–C). Source data are available online for this figure.

inhibition of the Hh pathway through knockdown of *ci* and *smo*, or overexpression of *ptc*, strongly suppressed tumor overgrowth, cell proliferation, and VNC invasion in *yki^{S168A}/Toll-6^{ACT}* tumors (Figs. 3D,E and EV3J–O). Conversely, hyperactivation of the Hh pathway through overexpression of a constitutively activated form of *ci* or *smo* was sufficient to synergize with *yki^{ACT}*, resulting in tumor overgrowth, enhanced proliferation, VNC invasion, and upregulation of Mmp1 (Figs. 3D–F and EV3J–O), phenocopying that of *yki^{ACT}/Toll-6^{ACT}* tumors. Together, these data indicate that Toll-6 functions through the Hh pathway to collaborate with activated *yki^{ACT}* in driving malignant tumor formation.

To further investigate whether activation of the Hh pathway is essential for the progression of QMARCM-induced *Qyki^{ACT}/scrib^{−/−}* tumors, we blocked Hh activation by co-depleting *ci* and *smo* or by expressing *ptc*. This intervention resulted in a notable suppression of tumor overgrowth and VNC invasion (Figs. 3G,H and EV3P). Interestingly, a previous study showed a physical interaction between Ci and Yki in *Drosophila* S2 cells (Li et al, 2015). Consistent with this, our immunostaining assay revealed a robust increase in Yki and Ci nuclear translocation in tumor clones co-expressing *yki^{S168A}* and *ci^{ACT}* (Fig. EV3Q), suggesting that activated *ci* and *yki* may physically interact to facilitate nuclear entry and promote tumor progression. Collectively, these findings indicate that Toll-6 activation synergizes with activated *yki* to promote tumor growth and invasion through activating the Hh pathway (Fig. 3I).

## Hh pathway acts upstream of JNK to drive tumor invasion

We have previously demonstrated that Toll-6 promotes organotropic metastasis by activating JNK signaling, a key regulator of cell migration (Mishra-Gorur et al, 2019). Given that Mmp1 can be transcriptionally activated by JNK (Uhlirova and Bohmann, 2006), we hypothesized that *yki^{ACT}/Toll-6^{ACT}* tumors upregulate Mmp1 and promote invasion in a JNK-signaling-dependent manner. We further explored the genetic interactions between the JNK and Hh pathways in the *yki^{ACT}/Toll-6^{ACT}* tumorigenic condition. We found that inhibiting JNK by expressing a dominant negative form of *bsk* (*bsk^{DN}*) did not affect *yki^{ACT}/Toll-6^{ACT}*-induced Ci induction (Fig. EV4A,B). Conversely, the knockdown of *ci* significantly suppressed Mmp1 upregulation (Fig. EV4C,D), suggesting that Ci genetically acts upstream of JNK signaling. Consistent with this, overexpression of *ci* is sufficient to upregulate Mmp1 expression (Fig. EV4E,F), whereas inhibition of *ci* significantly suppressed *Toll-6^{ACT}*-induced JNK activation, including Mmp1 and TRE-DsRed induction (Fig. EV4E–H). Moreover, the inhibition of JNK completely abolished the invasive migration to the VNC induced by *yki^{ACT}/Toll-6^{ACT}*, *Qyki^{ACT}/scrib^{−/−}*, and *yki^{ACT}/ci^{ACT}* (Fig. EV4I,J). Collectively, these data demonstrate that the Hh pathway acts upstream of JNK to drive tumor cell invasion.

## Toll-6^{ACT} activates Hh signaling by downregulating AP-2α

The accumulation of Smo on the cellular membrane is crucial for the activation of the Hh pathway (Jia and Jiang, 2022). Consistent with this, clonal Toll-6 activation significantly increased membrane accumulation of Smo (Fig. 4A). As endocytosis is the main process through which Smo is removed from the cell surface, and dysregulation of the endocytic pathway can lead to abnormal Smo accumulation (Li et al, 2012; Xia et al, 2012).

We further examined whether Toll-6 activates the Hh pathway through disrupting the Smo endocytosis process. Indeed, *Toll-6^{ACT}* clones exhibited impaired endocytosis, as indicated by Dextran endocytosis trafficking assay (Fig. 4B). Notably, our analysis of previous immunoprecipitation-mass spectrometry (IP-MS) data on Toll-6 has revealed a potential interaction between Toll-6 and the endocytic adaptor protein AP-2α (Kong et al, 2022). Indeed, proximity ligation assay (PLA) confirmed the close proximity between Toll-6 and AP-2α (Fig. 4C), and immuno-fluorescence staining revealed strong co-localization between Flag-tagged Toll-6 and HA-tagged AP-2α (Fig. EV5A,B). Interestingly, although Toll-6 physically interacts with AP-2α, Toll-6-activated clones demonstrated a significant downregulation of AP-2α expression when compared to wild-type clones (Figs. 4D and EV5C). Furthermore, overexpression of AP-2α effectively rescued the *Toll-6* activation-induced membrane accumulation of Smo (Fig. 4E,F), hindered the upregulation of Ci (Fig. 4G), and importantly, inhibited tumor overgrowth and VNC invasion caused by *yki^{S168A}/Toll-6^{ACT}* (Figs. 4H and EV5D,E). These data indicate AP-2α downregulation is essential for *Toll-6^{ACT}*-induced Hh activation and *yki^{S168A}/Toll-6^{ACT}*-induced tumor progression.

Next, we tested whether clonal mutation of *AP-2α* might activate Hh pathway and synergize with *yki^{ACT}* to induce malignant tumor formation. Notably, clones with *AP-2α* mutations exhibited elevated Ci expression (Fig. 4G), and *AP-2α^{−/−}/yki^{ACT}* tumors not only displayed robust overgrowth but also resulted in a significant increase in VNC invasion compared to *yki^{ACT}* clones (Figs. 4H and EV5D,E). Furthermore, *AP-2α* mutant clones showed decreased expression of the cell differentiation marker, Elav (Fig. EV5F), phenocopying *Toll-6^{ACT}* clones. Together, our data support the hypothesis that the activation of Toll-6 may impair normal Smo endocytosis by downregulating AP-2α. Consequently, this may result in the accumulation of Smo at the cell membrane and subsequent abnormal activation of the Hh pathway. Ultimately, Hh signaling activation synergizes with activated *yki* to promote tumor growth and invasion (Fig. 4I).

## Toll-6 promotes ubiquitination and degradation of AP-2α through Mib1

Next, we dissected the molecular mechanism by which activated Toll-6 downregulates AP-2α expression. Ubiquitination serves as a crucial posttranslational modification (PTM) in controlling substrate degradation. Intriguingly, upon re-examining Toll-6 IP-MS data, we identified Mind bomb 1 (Mib1), an E3 ubiquitin ligase, as a potential interacting protein of Toll-6 (Kong et al, 2022). Hence, we tested whether Toll-6 regulates AP-2α protein stability via Mib1. Firstly, we confirmed the physical association between ectopically expressed AP-2α, Mib1, and Toll-6 through co-immunoprecipitation (co-IP) in the S2 cells (Figs. 5A and EV6A). We also observed that both Toll-6 and Mib1 overexpression resulted in increased ubiquitination and subsequent degradation of AP-2α (Figs. 5B and EV6B). Furthermore, co-depletion of *mib1* rescued *Toll-6^{ACT}*-induced ubiquitination of AP-2α (Figs. 5C and EV6C). Additionally, employing MusiteDeep (Wang et al, 2020), a deep-learning framework-based PTM prediction tool, we identified three high-confidence lysine residues (K34, K290, and K857) within AP-2α as potential sites for ubiquitination (Fig. EV6D,E). Notably, mutation of K857, but not K34 or K290, significantly blocked Mib1-induced AP-2α ubiquitination (Fig. 5D), suggesting K857 of AP-2α is a target site of

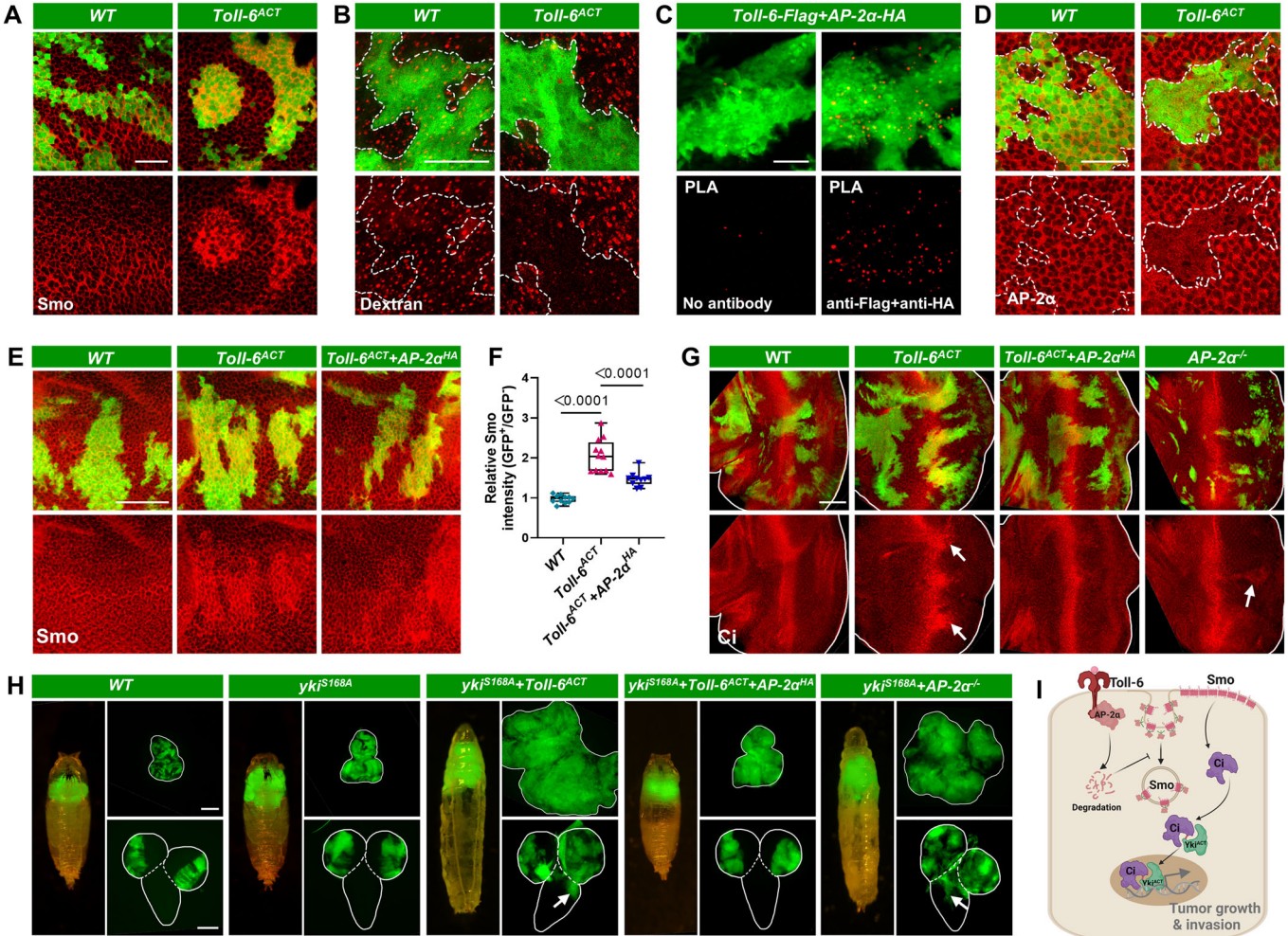

**Figure 4. Activated Toll-6 triggers Hedgehog signaling through AP-2α-dependent endocytosis defects.**

(A) E-A discs with GFP-labeled *WT* (left) and *Toll-6^ACT^* (right) clones were stained with anti-Smo antibody. (B) E-A discs with GFP-labeled *WT* (left) and *Toll-6^ACT^* (right) clones were subjected to the Dextran endocytosis assay, which revealed a reduction in Dextran uptake in *Toll-6^ACT^* clones. The white dashed lines mark the boundary of the GFP-positive clones. (C) Proximity ligation assay (PLA) using anti-FLAG and anti-HA antibodies in E-A discs shows that Toll-6-FLAG and AP-2α-HA are found in close proximity (<40 nm) (right). The negative control of the PLA experiment, where the primary antibody was absent, shows no signal (left). (D) E-A discs with GFP-labeled *WT* (left) and *Toll-6^ACT^* (right) clones were stained with anti-AP-2α antibody. The white dashed lines mark the boundary of the GFP-positive clones. (E) E-A discs with GFP-labeled *WT*, *Toll-6^ACT^*, and *Toll-6^ACT^/AP-2α^HA^* clones were stained with anti-Smo antibody. (F) Quantification of the relative intensity of Smo in E-A discs with indicated genotypes (from left to right, n = 11, 12, 10). (G) E-A discs with GFP-labeled *WT*, *Toll-6^ACT^*, *Toll-6^ACT^/AP-2α^HA^*, and *AP-2α^-/-^* clones were stained with anti-Ci antibody. The white lines outline the borders of the E-A discs. The arrows indicate increased Ci protein level. (H) Pupa/larva (left), E-A discs (right, top), and VNC (right, bottom) with GFP-labeled mosaic clones induced by ey-FLP/MARCM in *WT*, *yki^S168A^*, *yki^S168A^/Toll-6^ACT^*, *yki^S168A^/Toll-6^ACT^/AP-2α^HA^*, and *yki^S168A^/AP-2α^-/-^* genotypes. The white lines outline the borders of the E-A discs (right, top) and the VNC (right, bottom). The arrows indicate VNC invasion sites. (I) A schematic representation of Toll-6 activates Hedgehog signaling through AP-2α-dependent endocytosis defects. The *P* value of (F) was determined by unpaired nonparametric Mann–Whitney test. Exact *P* values are shown in the figures. The box plots of (F) boundaries represent the 25th (lower quartile) and 75th (upper quartile) percentiles, with the center line indicating the median, and the whiskers extend to the minimum and maximum values. Scale bars: 200 μm for (H); 50 μm for (E, G); 20 μm for (A–D). Source data are available online for this figure.

Mib1-mediated ubiquitination. Consistent with the in vitro data, clonal depletion of *mib1* in the E-A disc significantly reversed the *Toll-6^ACT^*-induced decrease in AP-2α (Fig. 5E) and the upregulation of Ci (Fig. 5F). Lastly, we investigated the requirement of *mib1* for tumor progression. Knockdown of *mib1* dramatically suppressed the tumor overgrowth and invasive migration caused by *yki^S168A^/Toll-6^ACT^* (Figs. 5G,H and EV6F) and *Qyki^ACT^/scrib^-/-^* (Fig. EV6G–I). These findings collectively suggest that Toll-6 facilitates the degradation of AP-2α by forming a complex with the E3 ubiquitin ligase Mib1. As a result, the Hedgehog pathway is

activated and collaborates with activated Yki to promote tumor progression.

## Tumor-derived Upds activates the JAK-STAT pathway in the fat body to upregulate Sd-dependent transcription of *spz5*

We further explored the mechanisms by which *spz5* is transcriptionally upregulated in the fat body. We performed bulk RNA-seq analyses to compare transcriptomic changes between *yki^S168A^* and

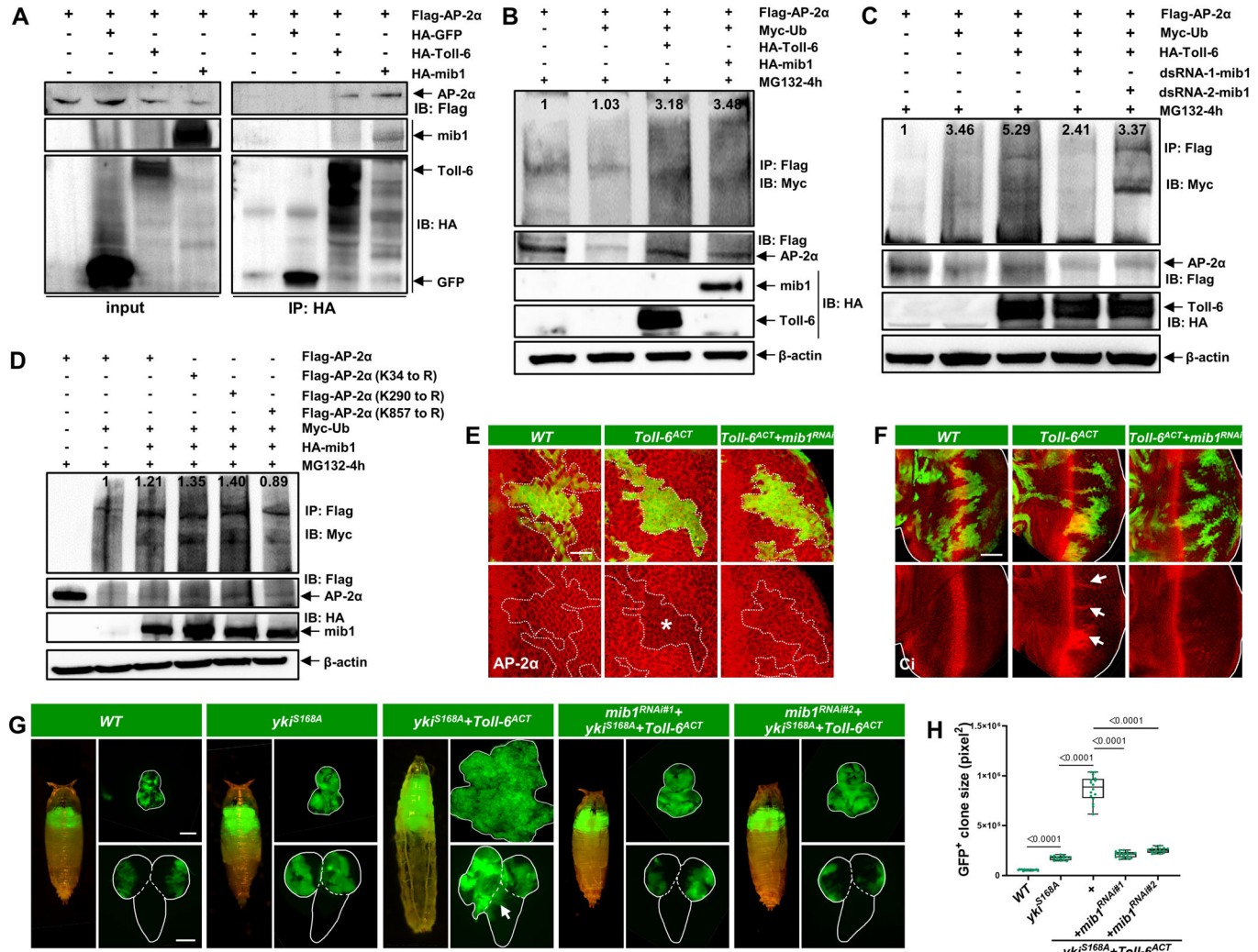

**Figure 5. Toll-6 facilitates AP-2α degradation through Mib1 ubiquitin ligase.**

(A) Western blotting showing results of proteins co-expressed in S2 cells. S2 cells transfected with Flag-tagged AP-2α, HA-tagged Toll-6, and HA-tagged Mib1. AP-2α physically interacts with Toll-6 and Mib1. (B) Western blotting showing results of ubiquitination experiments on proteins expressed in S2 cells. Overexpression of Toll-6 or Mib1 in S2 cells increases AP-2α ubiquitination. (C) Western blotting showing results of ubiquitination experiments on proteins expressed in S2 cells treated with dsRNA against Mib1. Mib1 is required for Toll-6-induced AP-2α ubiquitination in S2 cells. (D) Mib1 promotes ubiquitin-mediated protein degradation of AP-2α, and mutation of Lys-857 mitigates Mib1-induced AP-2α ubiquitination. (E, F) E-A discs with GFP-labeled *WT*, *Toll-6^ACT*, and *Toll-6^ACT/mib1^RNAi* clones were stained with anti-AP-2α antibody (E) or anti-Ci antibody (F). The white dashed lines in (E) mark the boundary of the GFP-positive clones, and the white lines in (F) outline the borders of the E-A discs. The asterisk in (E) indicates decreased AP-2α protein level, and the arrows in (F) indicate increased Ci protein level. (G) Pupa/larva (left), E-A discs (right, top), and VNC (right, bottom) with GFP-labeled mosaic clones induced by ey-FLP/MARCM in *WT*, *yki^S168A*, *yki^S168A/Toll-6^ACT*, *yki^S168A/Toll-6^ACT/mib1^RNAi#1*, and *yki^S168A/Toll-6^ACT/mib1^RNAi#2* genotypes. The white lines outline the borders of the E-A discs (right, top) and the VNC (right, bottom). The arrow indicates VNC invasion site. (H) Quantification of GFP-labeled clone size in E-A discs with the indicated genotypes (from left to right, n = 17, 15, 16, 18, 15). The P value of (H) was determined by unpaired nonparametric Mann–Whitney test. Exact P values are shown in the figures. The box plots of (H) boundaries represent the 25th (lower quartile) and 75th (upper quartile) percentiles, with the center line indicating the median, and the whiskers extend to the minimum and maximum values. Scale bars: 200 μm for (G); 50 μm for (F); 20 μm for (E). Source data are available online for this figure.

*yki^S168A/Toll-6^ACT* tumors. These analyses revealed a total of 16 significantly upregulated ligands in the *yki^S168A/Toll-6^ACT* tumor, among which the Unpaired (Upd) family of cytokines, *upd1*, *upd2*, and *upd3* being the top three, showing at least a 34-fold increase (Fig. EV7A; Dataset EV1). Consistently, we detected dramatic transcriptional upregulation of *upd1/2/3* in both *Qyki^ACT/scrib^−/−* and *yki^S168A/Toll-6^ACT* tumors (Figs. 6A and EV7B). Notably, we also observed transcriptional upregulation of *PDGF-* and *VEGF-related factors* (*Pvf1/2/3*) (Fig. EV7B), which are known to be secreted from

tumors to induce cachexia-like symptoms in *Drosophila* (Liu et al, 2022b; Song et al, 2019). Subsequently, we cell-autonomously inhibited these ligands and found that *upd1/2/3* inhibition significantly blocked the tumor overgrowth and invasion phenotype (Fig. 6B–D). In contrast, inhibiting *Pvf1/2/3* had no significant effect (Fig. EV7C–E).

We then evaluated whether tumor-secreted Upds (1/2/3) could be responsible for the increased expression of *spz5* in the distal fat bodies. Indeed, knockdown of *upd1/2/3*, but not *Pvfs*, significantly

impeded $Qyki^{ACT}/scrib^{-/-}$ tumor-induced upregulation of Spz5 in the fat bodies (Fig. 6E and EV7F,G). This suggests that tumor-derived Upds are essential for the systemic induction of Spz5 in distal fat bodies. The Upds family members are crucial ligands that

activate the Janus Kinase/Signal Transducers and Activators of Transcription (JAK/STAT) signaling pathway in *Drosophila*. This conserved signaling pathway plays a pivotal role in various developmental processes and pathological conditions, such as

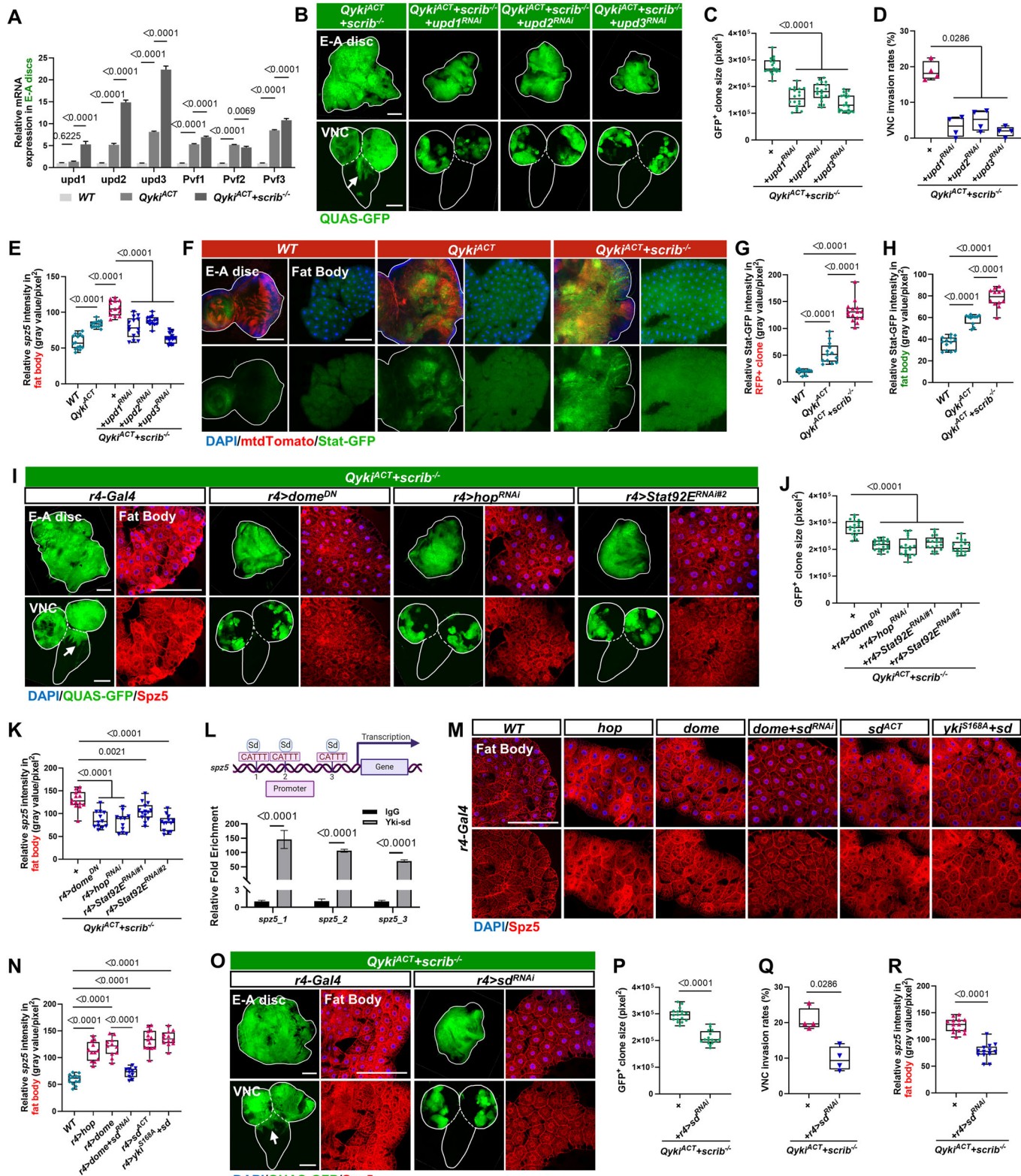

◄ **Figure 6. Tumor-derived Upds remotely promote Spz5 secretion by activating the JAK-STAT pathway in fat body.**

(A) Quantitative real-time PCR (qRT-PCR) to determine the mRNA levels of indicated genes in E-A discs with the *WT*, *Qyki^ACT^*, or *Qyki^ACT^/scrib^−/−^* genotypes (*n* = 3 independent experiments). (B) E-A discs (top) and VNC (bottom) with ey-FLP/QMARCM-induced GFP-labeled clones of *Qyki^ACT^/scrib^−/−^*, *Qyki^ACT^/scrib^−/−^/upd1^RNAi^*, *Qyki^ACT^/scrib^−/−^/upd2^RNAi^*, and *Qyki^ACT^/scrib^−/−^/upd3^RNAi^*. The white lines outline the borders of the E-A discs (top) and the VNC (bottom). The arrow indicates VNC invasion site. (C, D) Quantification of GFP-labeled clone size in E-A discs (C, from left to right, *n* = 17, 18, 19, 21), or percentage of clone invasion into VNC (D, total number from four independent experiments, from left to right, *n* = 203, 222, 226, 204) for the indicated genotypes. (E) Quantification of the relative intensity of Spz5 in fat bodies with indicated genotypes (from left to right, *n* = 14, 12, 13, 14, 13, 15). (F) E-A discs (left) and fat bodies (right) with mtdTomato-labeled clones induced by ey-FLP/QMARCM in *WT*, *Qyki^ACT^*, and *Qyki^ACT^/scrib^−/−^* genotypes were examined using the JAK-STAT signaling reporter Stat-GFP. The white lines outline the borders of the E-A discs. (G, H) Quantification of Stat-GFP in mtdTomato-labeled E-A discs (G, from left to right, *n* = 12, 13, 15), or fat bodies (H, from left to right, *n* = 13, 11, 12) with indicated genotypes. (I) E-A discs (left, top) and VNC (left, bottom) with GFP-labeled *Qyki^ACT^/scrib^−/−^* clones induced by ey-FLP/QMARCM, and expression of *WT*, *dome^DN^*, *hop^RNAi^*, and *Stat92E^RNAi#2^* in fat bodies under control of *r4-Gal4*; fat bodies (right) were stained with anti-Spz5 antibody. The white lines outline the borders of the E-A discs (left, top) and the VNC (left, bottom). The arrow indicates VNC invasion site. (J) Quantification of GFP-labeled clone size in E-A discs with the indicated genotypes (from left to right, *n* = 18, 20, 19, 17, 19). (K) Quantification of the relative intensity of Spz5 in fat bodies with indicated genotypes (from left to right, *n* = 14, 12, 11, 14, 13). (L) The schematic illustrates the promoter region (2 kb) and the transcription start site (TSS) of the spz5 locus. The potential sd binding motif (CATTT) and the target regions for qPCR enrichment are marked. *Drosophila* S2 cells co-transfected with HA-Sd and Yki-Myc were used to quantify the enrichment of Yki and Sd at these regions (*n* = 3). (M) Fat bodies expressing *WT*, *hop*, *dome*, *dome/sd^RNAi^*, *sd^ACT^*, and *yki^S168A^/sd* under the control of the *r4-Gal4* promoter were stained with anti-Spz5 antibody. (N) Quantification of the relative intensity of Spz5 in fat bodies with indicated genotypes (from left to right, *n* = 13, 12, 11, 10, 13, 12). (O) E-A discs (left, top) and VNC (left, bottom) with GFP-labeled *Qyki^ACT^/scrib^−/−^* clones induced by ey-FLP/QMARCM, and expression of *WT* or *sd^RNAi^* in fat bodies under the control of *r4-Gal4*; fat bodies (right) were stained with anti-Spz5 antibody. The white lines outline the borders of the E-A discs (left, top) and the VNC (left, bottom). The arrow indicates VNC invasion site. (P, Q) Quantification of GFP-labeled clone size in E-A discs (P, from left to right, *n* = 17, 16), or percentage of clone invasion into VNC (Q, total number from four independent experiments, from left to right, *n* = 216, 238) for the indicated genotypes. (R) Quantification of the relative intensity of Spz5 in fat bodies with indicated genotypes (from left to right, *n* = 14, 13). The *P* values of (A) were determined using one-way ANOVA with Tukey's multiple comparison test; the *P* values of (C, D, E, G, H, J, K, N, P, Q, R) were determined by unpaired nonparametric Mann–Whitney test; the *P* values of (L) were determined by unpaired two-tailed Student's *t* test. Exact *P* values are shown in the figures. The box plots of (C, D, E, G, H, J, K, N, P, Q, R) boundaries represent the 25th (lower quartile) and 75th (upper quartile) percentiles, with the center line indicating the median, and the whiskers extend to the minimum and maximum values. Scale bars: 200 μm for (B, F, I, M, O). DAPI 4′,6-diamidino-2-phenylindole. Source data are available online for this figure.

immune responses, tissue homeostasis, and tumorigenesis (Amoyel et al, 2014; Herrera and Bach, 2019). Upd ligands bind to their cognate receptors, Domeless (Dome), leading to the phosphorylation and activation of JAK. Activated JAK then phosphorylates STAT, which subsequently dimerizes and translocates to the nucleus to regulate target gene transcription. Notably, STAT activation is observed both cell-autonomously in tumors (Fig. 6F,G) and non-cell-autonomously in distal fat bodies (Fig. 6F,H). Furthermore, the knockdown of *upds* within the tumors inhibited STAT signaling activation in the fat bodies, as demonstrated by the transcriptional downregulation of multiple STAT target genes (Fig. EV7H). In addition, consistent with the cell-autonomous increase in JAK-STAT activity, inhibiting the JAK-STAT pathway by expressing a dominant negative form of *dome* (*dome^DN^*) within the tumors autonomously impeded tumor growth and invasion (Fig. EV7I–K).

Next, we dissected the role of the JAK-STAT pathway in tumor-bearing fat bodies using dual expression systems. Notably, genetic inhibition of JAK-STAT pathway specifically in the fat bodies, by expressing *dome^DN^*, or by knockdown of *hop* (JAK) or *Stat92E* (STAT), not only significantly suppressed the overgrowth and invasion of *Qyki^ACT^/scrib^−/−^* tumors (Figs. 6I,J and EV7L) but also impeded tumor-induced upregulation of Spz5 in these tissues (Figs. 6I,K and EV7M). Conversely, knockdown of *PDGF- and VEGF-receptor related* (*Pvr*), the receptor tyrosine kinase of Pvfs, within the fat bodies had no effect on tumor progression or *spz5* upregulation (Fig. EV7N–Q).

We then explored the mechanism by which *spz5* is transcriptionally upregulated in the tumor-bearing fat bodies. We failed to identify any potential binding sites of Stat92E on the 2 kb promoter regions of *spz5*, suggesting that activation of the JAK-STAT pathway may not directly regulate *spz5* transcription. JAK-STAT signaling activation can promote Yki-dependent hematopoietic cell proliferation (Anderson et al, 2017). Intriguingly, we identified

multiple putative binding motifs of Scalloped (Sd), the downstream transcription factor of Hippo signaling. Our cleavage under targets and tagmentation (CUT&Tag) analysis in *Drosophila* S2 cells confirmed that the Yki/Sd transcriptional complex directly binds to multiple sites within the *spz5* promoter region, indicating that the Hippo pathway directly regulates *spz5* transcription (Fig. 6L). In accordance with the putative genetic epistasis interaction between the JAK-STAT pathway and Hippo pathway, we observed that activation of JAK-STAT signaling under physiological conditions in the fat bodies significantly upregulates Spz5 expression in a Sd-dependent manner (Figs. 6M,N and EV7R). Conversely, ectopic expression of an activated form of Sd, or co-expression of Yki and Sd, is sufficient to upregulate Spz5 expression in the fat bodies (Fig. 6M,N and EV7R). In addition, several target genes of the Hippo pathway, including *Merlin* (*Mer*), *four-jointed* (*fj*), *Cyclin E* (*CycE*), *cactus* (*cact*), and *Myc* (Fig. EV7S), are also upregulated upon genetic activation of JAK-STAT signaling in the fat bodies (Fig. EV7T). More importantly, under tumor-induced pathological conditions, fat body-specific inhibition of the JAK-STAT signaling suppresses *Qyki^ACT^/scrib^−/−^*-induced upregulation of Hippo target genes (Fig. EV7U), while the knockdown of *sd* specifically in the fat bodies inhibits *Qyki^ACT^/scrib^−/−^*-induced tumor progression and Spz5 induction in the fat body (Figs. 6O–R and EV7V).

In summary, our data suggest that tumor-derived Upds activate the JAK-STAT signaling pathway in the distal fat bodies, which subsequently inactivates the Hippo pathway to directly upregulate *spz5* transcription, thereby systemically promoting the progression of distal *Qyki^ACT^/scrib^−/−^* tumors.

# Discussion

Traditionally, cancer research centered on intrinsic gene mutations as primary risk factors. Recent insights, however, emphasize the

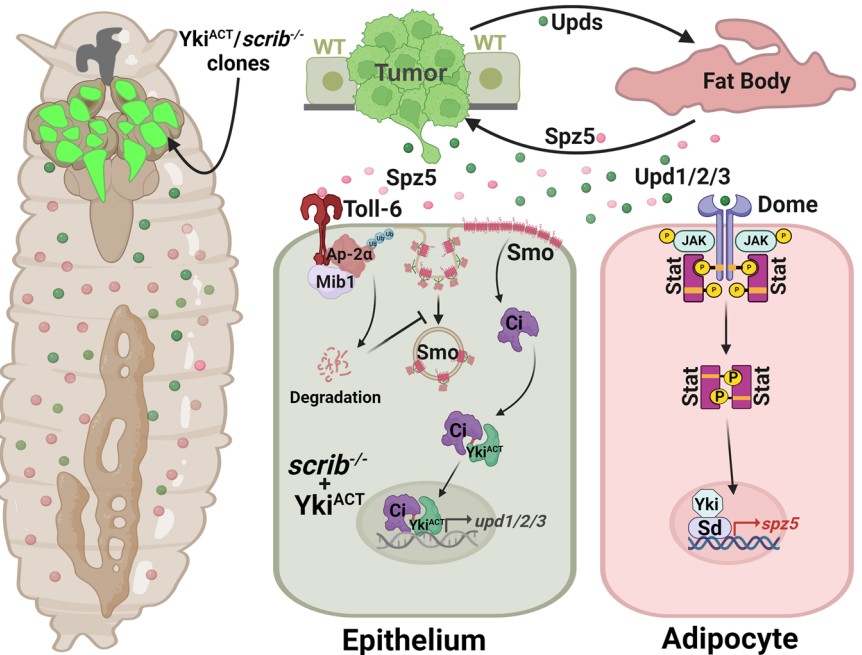

**Figure 7. Spz5-Toll-6 axis-mediated adipose-tumor crosstalk promotes tumor progression via Hedgehog pathway activation.**

Adipose tissue-secreted Spz5 ligand binds to epithelial tumor cell Toll-6 receptors, triggering the ubiquitination and degradation of the endocytosis adaptor protein AP-2α through the E3 ligase Mib1. Consequently, impaired AP-2α leads to the accumulation of Smo on the cell membrane, resulting in abnormal activation of the Hedgehog (Hh) signaling pathway. This activated Hh signaling pathway, in turn, synergizes with the oncogenic Yki to promote tumor growth and invasion. Simultaneously, tumor-derived Unpaired ligands (Upds) activate the JAK-STAT pathway in the adipocytes, which subsequently induces Hippo pathway-dependent upregulation of Spz5. This positive feedback loop between the epithelial tumor and the adipose tissue, mediated by Upds and Spz5, collectively facilitates tumor progression.

critical role of extrinsic elements, including adipose tissue, in cancer progression through their influence on growth factors, inflammation, and metabolism. Our study utilizes a refined dual-system *Drosophila* model to simultaneously target gene expression in both adipose and tumor tissues. This innovative method revealed that Spz5, an adipokine originating from the fat body, is an essential gene that remotely facilitates tumor progression. Furthermore, we demonstrate that the activation of the Spz5 receptor Toll-6 acts through the E3 ligase Mib1 to degrade AP-2α, resulting in Smo endocytosis defect that subsequently activates the Hedgehog pathway. This activation synergizes with activated Yki to drive malignant tumor progression in epithelial cells. In turn, the Upd1/2/3 derived from this tumor activates the JAK-STAT signaling pathway in the fat body, promoting the secretion of Spz5 from the fat body in a Hippo pathway-dependent manner (Fig. 7). Our study not only highlights the interorgan communications between adipose tissue and tumor, thereby presenting new perspectives on tumor biology but also provides a novel molecular insight into Hh pathway activation-mediated tumor progression.

Epidemiological studies robustly link excess adiposity to heightened risks of colorectal, pancreatic, liver, gastric, and breast cancers (Cascetta et al, 2018; Font-Burgada et al, 2016; Jamieson et al, 2011; Murphy et al, 2018; Renehan et al, 2015; Ruiz et al, 2023). Animal research reveals that visceral fat removal in high-fat-fed mice curtails colon and skin cancer incidence (Chakraborty et al, 2017). Yet, the precise mechanisms, especially identifying adipose-secreted tumor-promoting factors, remain elusive. Here, we introduce a *Drosophila* model leveraging the FLP-FRT, Gal4-

UAS, and QF-QUAS systems to probe adipose-tumor communication. We uncovered that Spz5, an adipose-secreted ligand, accelerates the growth and invasiveness of distant epithelial malignancies, suggesting its role as a novel adipokine in tumor progression. Numerous recent studies have highlighted the importance of using *Drosophila* as a model to elucidate tumor-host interactions (Bilder et al, 2021). Overexpression of the Yki oncogene in fly intestinal stem cells yields tumors secreting ImpL2, an insulin antagonist, causing cachexia-like symptoms (Newton et al, 2020). Similarly, Pvf1, secreted by *Drosophila* intestinal tumors, acts as a cachexia-inducing factor (Xu et al, 2023). Excessive Yki-activated tumors also secrete Ilp8, binding to Lgr3 on brain neurons, downregulating appetite-promoting neuropeptides, and inducing anorexia (Yeom et al, 2021). Interestingly, the bulk RNA-seq data demonstrate that Pvf1/2/3 are also upregulated within our tumor model. However, the in vivo genetic results indicate that knocking down *Pvf1/2/3* does not significantly inhibit the growth and invasion of *QykiACT/scrib⁻/⁻* tumors. Therefore, it is worthwhile to conduct future research to investigate whether these additionally upregulated ligands play other roles, such as in cachexia and anorexia.

Spz5, a member of the Spätzle family in *Drosophila*, acts as a neurotrophic factor and is processed by Furin to activate the Toll-6 receptor, a TLR family member, playing roles in development, immunity, and tissue homeostasis (McIlroy et al, 2013; Nonaka et al, 2018; Sutcliffe et al, 2013). Our previous study has shown that the fat body, which is central to innate immunity and epithelial surveillance, secretes Spz5 to induce Toll-6-dependent cell

clearance of *scrib* mutant cells. This is achieved by enhancing α-Spectrin-mediated tensional changes and subsequent Hippo pathway activation (Kong et al, 2022). These findings suggest a tumor-suppressive role of Toll-6. However, in the current study, we found that in *scrib* mutant cells with activated Yki, this Spz5-Toll-6 axis paradoxically accelerates tumor growth and invasion, indicating that activated Yki hijacks the tumor-suppressive role of Toll-6. Mechanistically, Toll-6 activation erroneously triggers the Hh signaling pathway, synergizing with activated Yki to foster malignant tumors. This complex dual role of the Spz5-Toll-6 axis in both tumor suppression and promotion underscores the fat body's multifaceted impact on tumor progression under varying conditions or stages. Further studies are warranted to elucidate the fat body's role across different tumor contexts.

Our study uncovers a novel interorgan communication axis between adipose tissue and tumors in *Drosophila*. In mammals, adipokines and the tumor microenvironment can similarly form a bidirectional positive feedback network through metabolic reprogramming (e.g., FFA exchange), inflammatory signaling (e.g., TNF-α/NF-κB pathways), or immune checkpoint regulation (e.g., PD-L1, CCL2) (Mukherjee et al, 2023; Mukherjee et al, 2022; Song et al, 2024; Wu et al, 2023; Wu et al, 2021). Future investigations should employ temporal multi-omics approaches to map the spatiotemporal dynamics of adipokine signatures during malignant progression, while developing precise targeting strategies to pharmacologically uncouple this oncogenic circuit. Given the evolutionary conservation of TLRs and the Hh signaling pathway, the mechanisms identified in *Drosophila* may be relevant to mammalian systems (Ji and Hoffmann, 2024). Our findings provide a foundational understanding of how adipose tissue might influence tumorigenesis in humans, setting the stage for future mammalian studies to further elucidate these mechanisms. The insights gained could potentially lead to novel therapeutic strategies targeting deregulated signaling pathways in cancer.

# Methods

### Reagents and tools table

| Reagent/resource | Reference or source | Identifier or catalog number |
| --- | --- | --- |
| **Experimental models** | | |
| *Drosophila* S2 cells | Gift from José Carlos Pastor-Pareja (Tsinghua University) | FBtc9000001 |
| *Drosophila* strains | This study | Table EV1 |
| **Antibodies** | | |
| Mouse monoclonal anti-Mmp1 | DSHB | Cat# 3A6B4, RRID: AB_579780 Cat# 3B8D12, RRID: AB_579781 Cat# 5H7B11, RRID: AB_579779 |
| Mouse monoclonal anti-β-gal | DSHB | Cat# 40-1a, RRID: AB_528100 |
| Mouse monoclonal anti-NICD | DSHB | Cat# C17.9C6, RRID: AB_528410 |
| Mouse monoclonal anti-NECD | DSHB | Cat# F461.3B, RRID: AB_528409 |

| Reagent/resource | Reference or source | Identifier or catalog number |
| --- | --- | --- |
| Mouse monoclonal anti-Hindsight | DSHB | Cat# 1G9, RRID: AB_528278 |
| Mouse monoclonal anti-Cut | DSHB | Cat# 2B10, RRID: AB_528186 |
| Mouse monoclonal anti-wg | DSHB | Cat# 4D4, RRID: AB_528512 |
| Mouse monoclonal anti-arm | DSHB | Cat# N2 7A1, RRID: AB_528089 |
| Mouse monoclonal anti-Smo | DSHB | Cat# 20C6, RRID: AB_528472 |
| Mouse monoclonal anti-Ptc | DSHB | Cat# Apa 1, RRID: AB_528441 |
| Rat monoclonal anti-Ci | DSHB | Cat# 2A1, RRID: AB_2109711 |
| Rat monoclonal anti-Elav | DSHB | Cat# 7E8A10, RRID: AB_528218 |
| Rat monoclonal anti-spi | DSHB | Cat# anti-Spitz, RRID: AB_528474 |
| Mouse monoclonal anti-DYKDDDDK-Tag | CST | Cat# 8146, RRID: AB_10950495 |
| Rabbit monoclonal anti-HA-Tag | CST | Cat# 3724, RRID: AB_1549585 |
| Rabbit monoclonal anti-pErk | CST | Cat# 4370S, RRID: AB_2315112 |
| Rabbit polyclonal anti-cleaved *Drosophila* Dcp-1 | CST | Cat# 9578, RRID: AB_2721060 |
| Rabbit polyclonal anti-phospho-histone H3 | CST | Cat# 9701, RRID: AB_331535 |
| Mouse anti-HRP | ABclonal | Cat# AS003, RRID: AB_2769851 |
| Rabbit anti-Myc-Tag | ABclonal | Cat# AE009, AB_2771925 |
| Mouse anti-Flag-Tag | Proteintech | Cat# 66008-4-Ig, RRID: AB_2918475 |
| Rabbit anti-HRP | Proteintech | Cat#SA00001-2, RRID: AB_2722564 |
| Rabbit anti-HA-Tag | Proteintech | Cat# 51064-2-AP, RRID: AB_11042321 |
| Mouse monoclonal anti-beta Actin | Abways | Cat# AB0061 |
| Rabbit polyclonal anti-Toll-6 | Xianjue Ma Lab | Kong et al, 2022 |
| Rabbit polyclonal anti-Spz5 | Xianjue Ma Lab | Kong et al, 2022 |
| Rabbit monoclonal anti-AP-2α | Gift from Marcos González-Gaitán (University of Geneva) | Gonzalez-Gaitan and Jackle, 1997 |
| Goat anti-Mouse IgG (H + L) Alexa Fluor™ Plus 555 | Invitrogen | Cat# A32727, RRID: AB_2633276 |
| Goat anti-Rabbit IgG (H + L) Alexa Fluor™ Plus 555 | Invitrogen | Cat# A32732, RRID: AB_2633281 |
| Goat anti-Rat IgG (H + L) Alexa Fluor™ Plus 555 | Invitrogen | Cat# A21434, RRID: AB_141733 |

| Reagent/resource | Reference or source | Identifier or catalog number |
|---|---|---|
| Goat anti-Rabbit IgG (H + L) Alexa Fluor™ Plus 647 | Invitrogen | Cat# A32733, RRID: AB_2633282 |
| **Oligonucleotides and other sequence-based reagents** | | |
| Oligonucleotides | This study | Table EV1 |
| **Chemicals, enzymes, and other reagents** | | |
| Antifade Mounting Medium with DAPI | Vector Laboratories | Cat# H-1800 |
| MG132 | Aladdin | Cat# M126521-100mg |
| Protein A/G Magnetic Beads | MedChemExpress | Cat# HY-K0202 |
| Anti-Flag Magnetic Beads | MedChemExpress | Cat# HY-K0207 |
| PVDF membrane | Merck Millipore | Cat# IPVH00010 |
| Effectene Transfection Reagent | Qiagen | Cat# 301427 |
| ESF 921 Insect Cell Culture Medium | Expression System | Cat# 96-001-01 |
| Fetal bovine serum | Cellmax | Cat# SA112 |
| Penicillin–streptomycin | Thermo Fisher Scientific | Cat# 15070063 |
| Alexa Fluor™ 555 Dextran | Invitrogen | Cat# D34679 |
| MEGAscript™ RNAi kit | Invitrogen | Cat# AM1626 |
| Taq Pro Universal SYBR qPCR Master Mix | Vazyme | Cat# Q712-02 |
| HiScript II 1st Strand cDNA Synthesis Kit | Vazyme | Cat# R211-01 |
| TRIzol Reagent | GlpBio | Cat# GK20008 |
| Duolink In situ PLA kits | Sigma-Aldrich | Cat# DUO92008-100RXN; DUO92002-100RXN; DUO92004-100RXN |
| **Software** | | |
| GraphPad Prism 9.0 | http://www.graphpad.com | RRID: SCR_002798 |
| ImageJ | https://imagej.net | RRID: SCR_003070 |
| Adobe Photoshop | https://www.adobe.com/products/photoshop.html | RRID: SCR_014199 |

## Methods and protocols

### Fly husbandry and genetics

*Drosophila* stocks and crosses were maintained on standard food at a temperature of 25 °C, unless otherwise specified. The composition of the standard cornmeal yeast food was as follows: 50 g of corn flour, 9 g of agar, 24.5 g of dry yeast, 7.25 g of white sugar, 30 g of brown sugar, 4.4 ml of propionic acid, 12.5 ml of ethanol, and 1.25 g of nipagin per liter. For a complete list of *Drosophila* lines used, please refer to Table EV1. In addition, detailed genotypes for each figure panel can be found in Table EV2.

### Immunofluorescence and imaging

Fluorescently labeled mitotic clones were generated in the eye discs following previously described procedures (Lee and Luo, 1999). Third-instar larval imaginal discs were dissected in cold PBS and fixed with 4% paraformaldehyde for 15 min at room temperature (RT) with gentle shaking, followed by three washes with PBS containing 0.1% Triton X-100 (PBST). The samples were then blocked in PBS with 5% normal goat serum for 1 h at RT and subsequently incubated with primary antibodies overnight at 4 °C. After washing with PBST; all samples were incubated with secondary antibodies for 2 h at RT. Following PBST washes, the samples were mounted with a DAPI-containing medium and imaged using a Nikon A1R confocal microscope (inverted) equipped with ×40 and ×60 oil objectives or a Zeiss Axio Observer with ApoTome.2. The acquired images were processed using ImageJ and Photoshop CS5 (Adobe) software for image merging and resizing.

For quantification of immunofluorescence staining intensity, more than 8 randomly selected discs were used. Fiji/ImageJ software was employed for automated quantitation of fluorescent intensity.

### Clone size measurement

Using fluorescent microscopy, capture images of *Drosophila* larvae imaginal discs with GFP or RFP-labeled cell clones. Open the desired images in ImageJ software for analysis. First, outline and record the entire area of the adult eye disc based on the DAPI signal that outlines the disc. Then, set a threshold to quantify and record the area occupied by GFP or RFP. To calculate the relative area of GFP or RFP within the entire disc, import the experimental data into Microsoft Excel software for further analysis. Statistical analysis and graphing can be performed by importing the experimental and control group data into GraphPad Prism software.

### Quantification of in vivo tumor invasion

*ey-FLP*-induced MARCM or QMARCM clones not only appear in the eye-antennal imaginal discs but also expressed in the brain lobes under normal conditions. However, these clones are not observed in the adjacent ventral nerve cord (VNC) under normal conditions. Therefore, in *Drosophila* tumor-related experiments, the VNC is often examined to assess the tumor's invasive capability, particularly in the context of tumor generated in the eye-antennal imaginal discs (Pagliarini and Xu, 2003). First, we utilized the MARCM or QMARCM technique to induce the formation of fluorescently labeled mosaic clones of distinct genotypes in the eye-antennal imaginal discs of *Drosophila* larvae. Second, we dissected the VNC from these larvae and performed DAPI staining to distinguish between the brain lobes and the VNC. We conducted four independent technical replicate experiments, and for each experiment, we calculated the ratio of VNCs that exhibited fluorescently labeled invasive clones, with at least 20 larvae per experiment. Consequently, we included at least 80 larvae for each genotype.

### Dextran endocytosis assay

Late third-instar larvae of the corresponding genotypes were dissected in M3 medium, and the imaginal discs were carefully

isolated. They were then placed in 0.5 mM Dextran (Invitrogen, Alexa Fluor™ 555, D34679) and incubated at 25 °C for 10 min. The tissues were washed five times with ice-cold M3 medium, each time for 2 min. Subsequently, they were fixed in 4% paraformaldehyde at room temperature for 20 min. After three washes with pre-chilled M3 medium, each for 5 min, the samples were mounted and imaged using a confocal microscope.

### Duolink in situ proximity ligation assay (PLA)

The PLA (Proximity Ligation Assay) was performed using the Duolink PLA Kit (Sigma), following previously described methods (Lindehell et al, 2015). Drosophila eye-antennal discs with corresponding genotypes were dissected in phosphate-buffered saline (PBS) at room temperature (RT) and fixed in freshly prepared 4% formaldehyde in PBS for 20 min. They were then washed three times with PBST (PBS buffer containing 0.1% Triton X-100) and blocked in 1% goat serum for 1 h. Subsequently, the discs were incubated overnight at 4 °C with the appropriate primary antibodies, followed by three washes with PBST. The remaining steps were conducted in the dark. The discs were incubated for 1 h at 37 °C with PLA probes diluted in 1% goat serum, followed by ligation at 37 °C for 30 min and amplification at 37 °C for 100 min. After the final wash, the discs were mounted in Duolink mounting media with DAPI.

### RNA extraction and real-time RT-PCR

Total RNAs extracted from third-instar larvae. Briefly, larvae were lysed in TRIzol and RNA extraction was performed according to the manufacturer's instructions. Total RNA was reversely transcribed into cDNA with HiScript II 1st Strand cDNA Synthesis Kit. The cDNA samples were stored at −20 °C. Quantitative real-time PCR (qRT-PCR) analysis was performed using Taq Pro Universal SYBR qPCR Master Mix with Jena Qtower384G Real-Time PCR System.

### RNA interference

Sample were according to the MEGAscript™ RNAi kit manufacturer's protocols. The genes expression level was analyzed by qRT-PCR. The sequence information of dsRNAs was provided in Table EV1.

### Drosophila cell culture

S2 cells were cultured in ESF 921 Insect Cell Culture Medium with 10% fetal bovine serum and 1% penicillin–streptomycin at 28 °C.

### Cell transfection

Plasmid transfection and double-stranded RNA (dsRNA) were transfection of S2 cells was performed using Effectene Transfection Reagent according to the manufacturer's instructions and harvested after an additional 72 h. The plasmid AP-2α were constructed by PCR into the pUAST-3XFlag vector. The plasmids mib1, AP-2α and Toll-6 were constructed by PCR into the pUAST-3XHA vector. All constructs were verified by DNA sequencing. pAc-6Myc-Ub, pUAST-3XFlag-GFP, and pUAST-3XHA-GFP were gifts from Xing Wang, China Agricultural University, Beijing.

### Immunoprecipitation and western blotting

For co-IP, harvested cells were washed with 1 ml cold PBS, Cells were lysed with NP-40 buffer (50 mM Tris (pH 7.4), 150 mM NaCl,

1% NP-40, sodium pyrophosphate, β-glycerophosphate, sodium orthovanadate, sodium fluoride, EDTA and leupeptin) supplemented with 1 mM PMSF. The samples were rotated for 30 min at 4 °C, and lysates were centrifuged at 13,000 rpm for 13 min. The lysate supernatant was incubated with primary antibody overnight at 4 °C; then pre-washed Pierce™ Protein A/G Magnetic Beads were added to the lysate and antibody solution for 3 h at 4 °C, the beads were separated by a magnetic separation rack. After that, treated with 5× sample loading buffer containing 0.1 M DTT and heated to 95 °C for 7 min. Separation of proteins by SDS-PAGE, proteins were transferred to PVDF membrane which was blocked with 5% milk-TBST for 1 h, incubated in primary antibody overnight at 4 °C, washed with TBST, incubated with secondary antibody at room temperature for 1 h, and washed by TBST three times for visualization. For the ubiquitination assay, 72 h after transfection, 25 μM MG132 was added to transfected cells 4 hr before harvesting cells. AP-2α ubiquitination modification site prediction through MusiteDeep (the threshold value is set to 0.45).

### Cut&Run and data analysis

Cut&Run experiments were conducted using the Hyperactive pG-MNase CUT&RUN Assay Kit for PCR/qPCR (Vazyme, HD101) in accordance with the manufacturer's protocols. In total, $1 \times 10^5$ Drosophila S2 cells co-transfected with HA-tagged Sd (HA-Sd) and myc-tagged Yki (Yki-myc) were collected for each sample. The cells were then gently resuspended in the cell wash buffer and incubated with concanavalin A (ConA) coated magnetic beads pre-washed with ConA binding buffer. The cells coupled with ConA beads were collected and incubated with primary antibodies (Rabbit IgG, anti-HA tag, or anti-Myc tag) at 4 °C overnight. On the following day, the pG-MNase enzyme was incubated with the ConA beads coupled cells at 4 °C for 1 h. The ConA beads coupled cells were then collected, and the standard tagmentation and amplification procedures were performed as detailed in the kit instructions. The enrichment of DNA sequences was quantified using qPCR with the primers listed in Table EV1.

### Bulk RNA-seq

A total of 200 eye-antenna-disc tumors from $yki^{S168A}$ (control group, 4 replicates) and $yki^{S168A}/Toll-6^{ACT}$ (case group, 4 replicates) were harvested at the late third-instar larval stage for total RNA extraction using Trizol (Invitrogen, Carlsbad, CA) according to the manufacturer's instructions. The mRNA library construction was performed as previously described (Liu et al, 2022a). The mRNA libraries were amplified with phi29 to create DNA nanoballs (DNBs), which were subsequently loaded onto a patterned nanoarray, and single-end 50-base reads were generated using the BGIseq500 platform (BGI-Shenzhen, China).

### Bulk RNA-seq data analysis

The transcriptomic data were processed through a standardized pipeline. Data quality control and read statistics were performed using FastQC v0.11.8 (https://github.com/s-andrews/FastQC/releases/tag/v0.11.8). High-quality reads were mapped to the Drosophila melanogaster reference genome (Drosophila_melanogaster.BDGP6.32.108) using Hisat2 (Kim et al, 2019). Gene feature counting was conducted using HTSeq 84 with default settings (Anders et al, 2015). Differential gene expression analysis and further data exploration were performed in R (version 4.2.0).

Differentially expressed genes (DEGs) were identified using the "RLE" method of edgeR (Robinson et al, 2010), meeting the criteria of |Fold-Change| > 1.5 and false discovery rate (FDR) < 0.05. Over-representation analyses (ORA) were conducted using the cluster-Profiler package (Yu et al, 2012), and DEGs were annotated against terms in the Gene Ontology (GO) consortium and the Kyoto Encyclopedia of Genes and Genomes (KEGG) database. Relevant results were visualized using the pheatmap (v1.0.12), RColorBrewer (v1.1-3), and ggplot2 (v3.4.4) packages.

### Graphics

Panels in Figs. 1I, P, 2H, 3I, 4I, 7, EV1C, and synopsis were created with BioRender.com.

### Quantification and statistical analysis

All statistical analyses were conducted using GraphPad Prism 9.0 software. ImageJ was utilized to quantify the relative area of mosaic clones. Unless otherwise indicated, comparisons between two genotypes/conditions were analyzed with unpaired nonparametric Mann–Whitney tests for nonparametric data. Unpaired two-tailed Student's $t$ test or ordinary one-way ANOVA test with Tukey's multiple comparisons test for parametric data. $P$ value < 0.05 was considered statistically significant.

## Data availability

All sequencing data of this study are deposited in the National Center for Biotechnology Information Sequence Read Archive (SRA) with the accession number BioProject: PRJNA1210978.

The source data of this paper are collected in the following database record: biostudies:S-SCDT-10_1038-S44318-025-00489-y.

## Peer review information

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

## Acknowledgements

The authors thank Tian Xu, Lei Xue, Zizhang Zhou, Xing Wang, Marcos Gonzalez-Gaitan, and Jose C. Pastor-Pareja for the generous sharing of fly stocks and reagents, and the Tsinghua Fly Center, BDSC, VDRC, the NIG-Fly Stock Center, and DSHB for stocks and antibodies. XM is supported by startup funds from Westlake University and the National Natural Science Foundation of China (32170824, 32322027), HRHI program (1011103360222B1) of Westlake Laboratory of Life Sciences and Biomedicine, and "Pioneer" and "Leading Goose" R&D Program of Zhejiang (2024SSYS0034). LH is supported by the National Natural Science Foundation of China (32070750) and the startup funds from USTC. DK is supported by the National Natural Science Foundation of China (32400588), Shandong Provincial Natural Science Foundation (ZR2024QH197), Postdoctoral Fellowship Program of CPSF (GZC20231456), and China Postdoctoral Science Foundation (CN2023M742079).

## Author contributions

**Du Kong**: Conceptualization; Data curation; Formal analysis; Funding acquisition; Investigation; Writing—original draft; Writing—review and editing. **Xiaoqin Li**: Investigation. **Sihua Zhao**: Investigation. **Chenliang Wang**: Investigation. **Zixin Cai**: Investigation. **Sha Song**: Investigation. **Yifan Guo**: Investigation. **Xiaoyu Kuang**: Investigation. **Xianping Wang**: Investigation. **Wenhan Liu**: Resources. **Peng Liu**: Investigation. **Xiaowei Guo**: Investigation. **Wenyan Xu**: Investigation. **Yirong Wang**: Investigation. **Bin Zhao**: Investigation. **Bin Jin**: Supervision; Validation. **Li He**: Supervision; Funding acquisition; Validation. **Xianjue Ma**: Conceptualization; Resources; Supervision; Funding acquisition; Validation; Project administration; Writing—review and editing.

Source data underlying figure panels in this paper may have individual authorship assigned. Where available, figure panel/source data authorship is listed in the following database record: biostudies:S-SCDT-10_1038-S44318-025-00489-y.

## Disclosure and competing interests statement

The authors declare no competing interests.

# Expanded View Figures

**Figure EV1. Fat body-derived Spz5 is essential for *Qyki^{ACT}/scrib^{-/-}* tumor progression in eye-antennal discs.**

(**A**) Ventral nerve cord (VNC) with GFP-labeled *WT*, *Qyki^{ACT}*, and *Qyki^{ACT}/scrib^{-/-}* clones induced by ey-FLP/QMARCM, stained with anti-Mmp1 antibody. (**B**) Eye-antennal discs (E-A discs, top) and VNC (bottom) with GFP and RFP double-labeled clones, respectively, by QF/QUAS and Gal4/UAS system. (**C**) Cartoon illustrating the *Drosophila* tumor interorgan communication model between fat body or hemocytes and E-A discs. The QMARCM system generates *Qyki^{ACT}/scrib^{-/-}* tumor in the larval E-A discs, whereas the Gal4/UAS system expresses UAS-controlled genes in either fat body or hemocytes. (**D**) E-A discs (top) and VNC (bottom) with GFP-labeled *Qyki^{ACT}/scrib^{-/-}* clones induced by ey-FLP/QMARCM, and expression of *WT*, *spz5^{RNAi#1}*, and *spz5^{RNAi#2}* in fat body under control of *r4-Gal4*, the arrow indicates VNC invasion site. (**E, F**) Quantification of GFP-labeled clone size in E-A discs (**E**, from left to right, $n = 19, 25, 23$), or percentage of clone invasion into VNC (**F**, total number from four independent experiments, from left to right, $n = 110, 127, 113$) for the indicated genotypes. (**G**) E-A discs (top) and VNC (bottom) with GFP-labeled *Qyki^{ACT}/scrib^{-/-}* clones induced by ey-FLP/QMARCM, and expression of *WT*, *spz5^{RNAi#1}*, and *spz5^{RNAi#2}* in hemocytes under control of *He-Gal4*, the arrows indicate VNC invasion sites. (**H, I**) Quantification of GFP-labeled clone size in E-A discs (**H**, from left to right, $n = 20, 24, 22$), or percentage of clone invasion into VNC (**I**, total number from four independent experiments, from left to right, $n = 119, 113, 139$) for the indicated genotypes. The *P* values of (**E, H, I**) were determined by unpaired nonparametric Mann–Whitney test. Exact *P* values are shown in the figures. The box plots of (**E, F, H, I**) boundaries represent the 25th (lower quartile) and 75th (upper quartile) percentiles, with the center line indicating the median, and the whiskers extend to the minimum and maximum values. Scale bars: 200 μm for (**A, B, D, G**). DAPI 4′,6-diamidino-2-phenylindole.

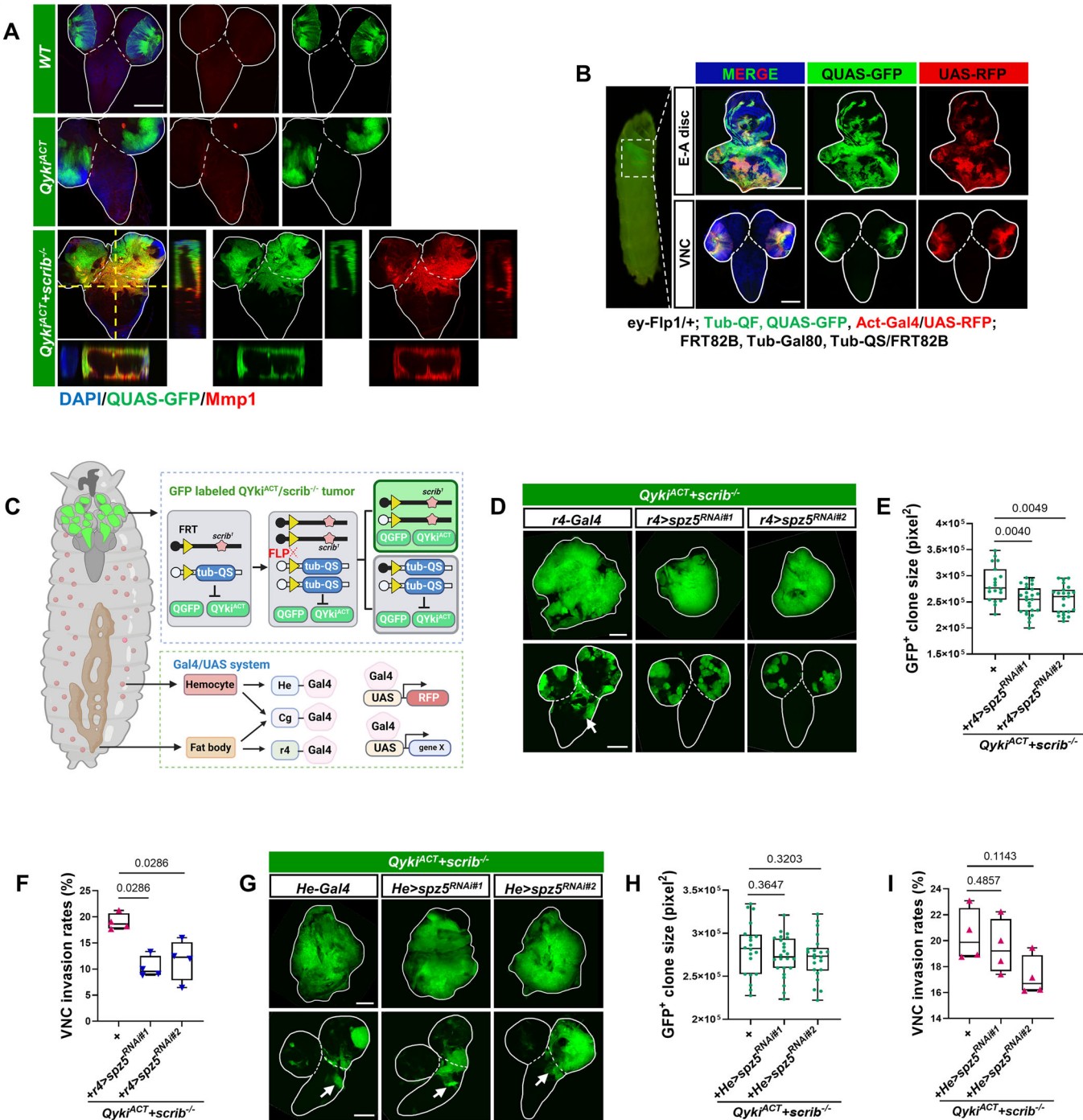

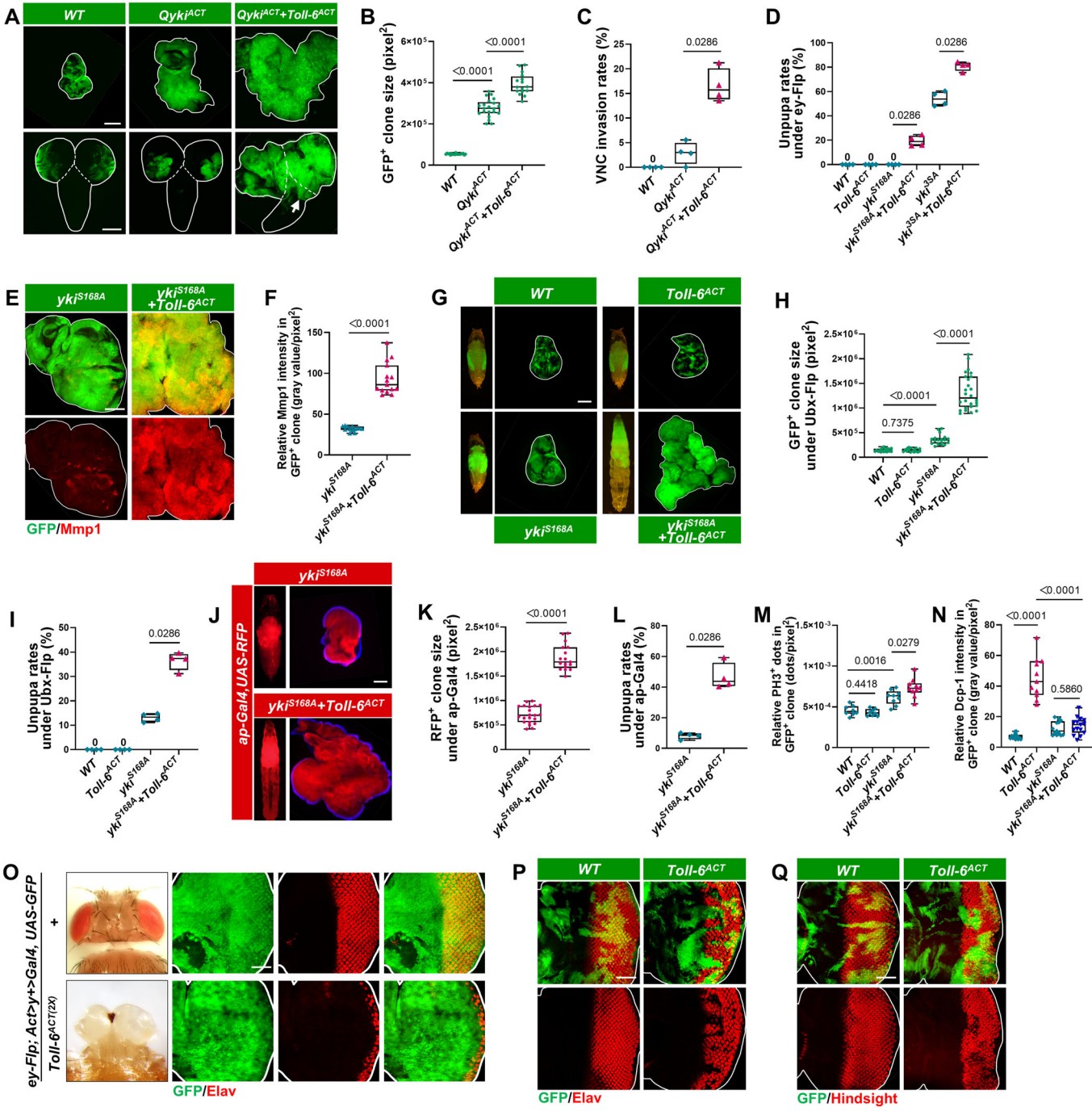

◄

**Figure EV2. Activated Toll-6 promotes Yki-driven tumor malignancy.**

(A) E-A discs (top) and VNC (bottom) with GFP-labeled clones induced by ey-FLP/QMARCM for *WT*, *Qyki^ACT^*, and *Qyki^ACT^/Toll-6^ACT^*, the arrow indicates VNC invasion site. (B, C) Quantification of GFP-labeled clone size in E-A discs (B, from left to right, $n = 19, 21, 16$), or percentage of clone invasion into VNC (C, total number from four independent experiments, from left to right, $n = 116, 134, 140$) for the indicated genotypes. (D) Quantification of *Drosophila* larval non-pupation rates with indicated genotypes (total number from four independent experiments, from left to right, $n = 224, 253, 204, 205, 226, 226$). (E) E-A discs with GFP-labeled *yki^S168A^* and *yki^S168A^/Toll-6^ACT^* clones were stained with anti-Mmp1 antibody. (F) Quantification of the relative Mmp1 intensity in clones with indicated genotypes (from left to right, $n = 13, 15$). (G) Pupa/larva (left) and wing imaginal discs (right) with GFP-labeled Ubx-FLP/MARCM-induced mosaic clones in *WT*, *Toll-6^ACT^*, *yki^S168A^*, and *yki^S168A^/Toll-6^ACT^* genotypes. (H, I) Quantification of GFP-labeled clone size in wing imaginal discs (H, from left to right, $n = 19, 16, 18, 26$), or percentage of *Drosophila* larval un-pupation rates (I, total number from four independent experiments, from left to right, $n = 186, 205, 220, 213$) for the indicated genotypes. (J) RFP-labeled pupa/larva (left) and wing imaginal discs (right) expressing *yki^S168A^* and *yki^S168A^/Toll-6^ACT^* under control of ap-Gal4. (K, L) Quantification of RFP-labeled wing imaginal disc size (K, from left to right, $n = 21, 17$), or percentage of *Drosophila* larval un-pupation rates (L, total number from four independent experiments, from left to right, $n = 234, 228$) for the indicated genotypes. (M) Quantification of PH3-positive dots in GFP-labeled clones with indicated genotypes (from left to right, $n = 8, 8, 9, 10$). (N) Quantification of Dcp-1 intensity in GFP-labeled clones with indicated genotypes (from left to right, $n = 14, 11, 13, 15$). (O) Adult head (left) and E-A discs (right) stained with anti-Elav antibody with indicated genotypes. (P, Q) E-A discs with GFP-labeled clones induced by ey-FLP/MARCM for *WT* and *Toll-6^ACT^* were stained with anti-Elav antibody (P) and anti-Hindsight antibody (Q). The P values of (B, C, D, F, H, I, K, L, M, N) were determined by unpaired nonparametric Mann–Whitney test. Exact P values are shown in the figures. The box plots of (B, C, D, F, H, I, K, L, M, N) boundaries represent the 25th (lower quartile) and 75th (upper quartile) percentiles, with the center line indicating the median, and the whiskers extend to the minimum and maximum values. Scale bars: 200 μm for (A, G, J); 100 μm for (E); 50 μm for (O–Q).

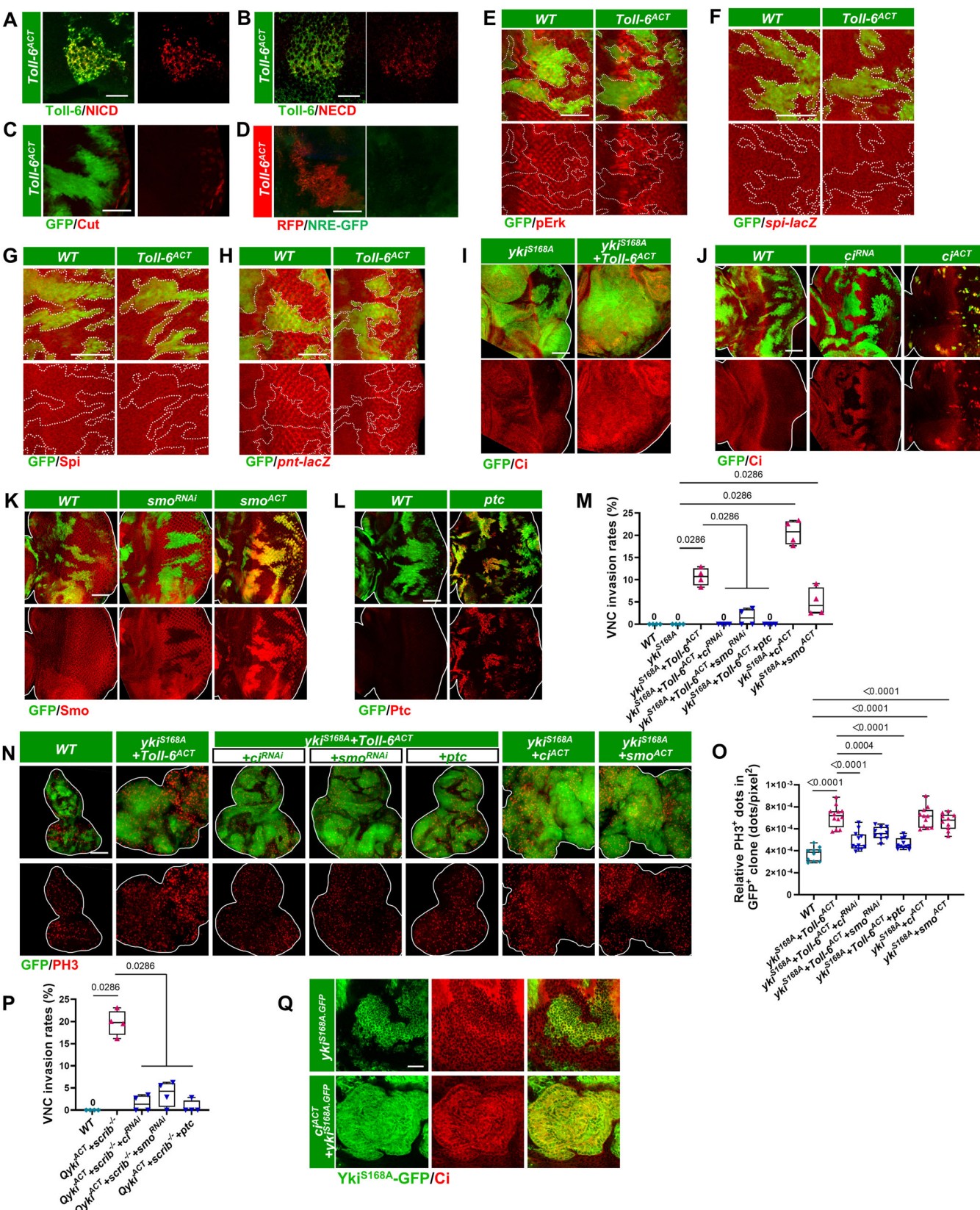

**Figure EV3.  Cooperation between Hedgehog activation and oncogenic Yki in driving tumorigenesis.**

(A) E-A discs with $Toll\text{-}6^{ACT}$ clones, co-immunostained with anti-Toll-6 and anti-NICD antibodies. (B) E-A discs with $Toll\text{-}6^{ACT}$ clones, co-immunostained with anti-Toll-6 and anti-NECD antibodies. (C) E-A discs with GFP-labeled $Toll\text{-}6^{ACT}$ clones were stained with anti-Cut antibody. (D) E-A discs with RFP-labeled $Toll\text{-}6^{ACT}$ clones were examined using the Notch signaling indicator NRE-GFP. (E) E-A discs with GFP-labeled $WT$ and $Toll\text{-}6^{ACT}$ clones were stained with anti-pErk antibody. (F) E-A discs with GFP-labeled $WT$ and $Toll\text{-}6^{ACT}$ clones were stained with anti-β-galactosidase antibody to label $spi$ transcription. (G) E-A discs with GFP-labeled $WT$ and $Toll\text{-}6^{ACT}$ clones were stained with anti-Spi antibody. (H) E-A discs with GFP-labeled $WT$ and $Toll\text{-}6^{ACT}$ clones were stained with anti-β-galactosidase antibody to label $pnt$ transcription. (I) E-A discs with GFP-labeled $yki^{S168A}$ and $yki^{S168A}/Toll\text{-}6^{ACT}$ clones were stained with anti-Ci antibody. (J) E-A discs with GFP-labeled $WT$, $ci^{RNAi}$, and $ci^{ACT}$ clones were stained with anti-Ci antibody. (K) E-A discs with GFP-labeled $WT$, $smo^{RNAi}$, and $smo^{ACT}$ clones were stained with anti-Smo antibody. (L) E-A discs with GFP-labeled $WT$ and $ptc$ overexpression clones were stained with anti-Ptc antibody. (M) Quantification of the percentage of clone invasion into VNC with indicated genotypes (total number from four independent experiments, from left to right, $n = 98, 110, 142, 124, 132, 85, 132, 146$), as related to Fig. 3D. (N) E-A discs with GFP-labeled $WT$, $yki^{S168A}/Toll\text{-}6^{ACT}$, $yki^{S168A}/Toll\text{-}6^{ACT}/ci^{RNAi}$, $yki^{S168A}/Toll\text{-}6^{ACT}/smo^{RNAi}$, $yki^{S168A}/Toll\text{-}6^{ACT}/ptc$, $yki^{S168A}/ci^{ACT}$, and $yki^{S168A}/smo^{ACT}$ clones were stained with anti-phospho-histone H3 (PH3) antibody. (O) Quantification of the number of PH3-positive cells in GFP-labeled clones with indicated genotypes (from left to right, $n = 10, 12, 12, 10, 11, 11, 10$). (P) Quantification of the percentage of clone invasion into VNC with indicated genotypes (total number from four independent experiments, from left to right, $n = 93, 146, 128, 130, 115$), as related to Fig. 3G. (Q) E-A discs with GFP-fused $yki^{S168A.GFP}$ (top) and $yki^{S168A.GFP}/ci^{ACT}$ (bottom) clones were stained with anti-Ci antibody. The $P$ values of (M, O, P) were determined by unpaired nonparametric Mann–Whitney test. Exact $P$ values are shown in the figures. The box plots of (M, O, P) boundaries represent the 25th (lower quartile) and 75th (upper quartile) percentiles, with the center line indicating the median, and the whiskers extend to the minimum and maximum values. Scale bars: 100 μm for (N); 50 μm for (C–L); 20 μm for (A, B, Q).

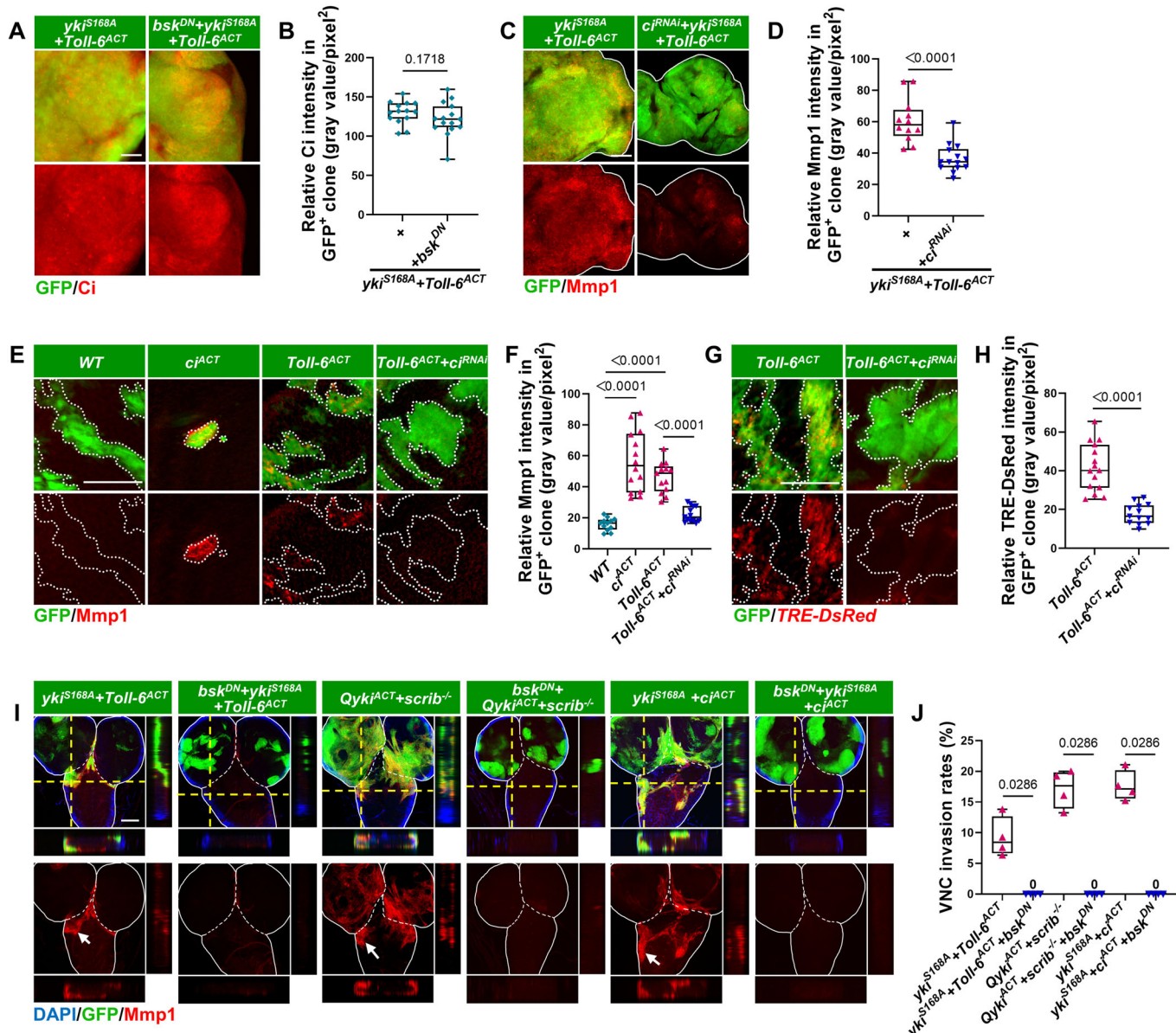

**Figure EV4. Hedgehog activation promotes JNK-dependent tumor invasion.**

(A) E-A discs with GFP-labeled *yki^{S168A}/Toll-6^{ACT}* and *yki^{S168A}/Toll-6^{ACT}/bsk^{DN}* clones were stained with anti-Ci antibody. (B) Quantification of the relative Ci intensity in GFP-labeled clones with indicated genotypes (from left to right, n = 14, 15). (C) E-A discs with GFP-labeled *yki^{S168A}/Toll-6^{ACT}* and *yki^{S168A}/Toll-6^{ACT}/ci^{RNAi}* clones were stained with anti-Mmp1 antibody. (D) Quantification of the relative Mmp1 intensity in GFP-labeled clones with indicated genotypes (from left to right, n = 12, 14). (E) E-A discs with GFP-labeled *WT*, *ci^{ACT}*, *Toll-6^{ACT}*, and *Toll-6^{ACT}/ci^{RNAi}* clones were stained with anti-Mmp1 antibody. (F) Quantification of the relative Mmp1 intensity in GFP-labeled clones with indicated genotypes (from left to right, n = 12, 14, 13, 13). (G) E-A discs with GFP-labeled *Toll-6^{ACT}* and *Toll-6^{ACT}/ci^{RNAi}* clones expressed the JNK reporter *TRE-DsRed*. (H) Quantification of the relative *TRE-DsRed* intensity in GFP-labeled clones with indicated genotypes (from left to right, n = 15, 13). (I) z-stack confocal images of GFP-labeled *yki^{S168A}/Toll-6^{ACT}*, *yki^{S168A}/Toll-6^{ACT}/bsk^{DN}*, *Qyki^{ACT}/scrib^{−/−}*, *Qyki^{ACT}/scrib^{−/−}/bsk^{DN}*, *yki^{S168A}/ci^{ACT}*, and *yki^{S168A}/ci^{ACT}/bsk^{DN}* clones were stained with anti-Mmp1 antibody. (J) Quantification of the percentage of clone invasion into VNC with indicated genotypes (total number from four independent experiments, from left to right, n = 241, 211, 231, 229, 214, 218). The P values of (B, D, F, H, J) were determined by unpaired nonparametric Mann–Whitney test. Exact P values are shown in the figures. The box plots of (B, D, F, H, J) boundaries represent the 25th (lower quartile) and 75th (upper quartile) percentiles, with the center line indicating the median, and the whiskers extend to the minimum and maximum values. Scale bars: 100 μm for (C) and I; 50 μm for (A, E, G).

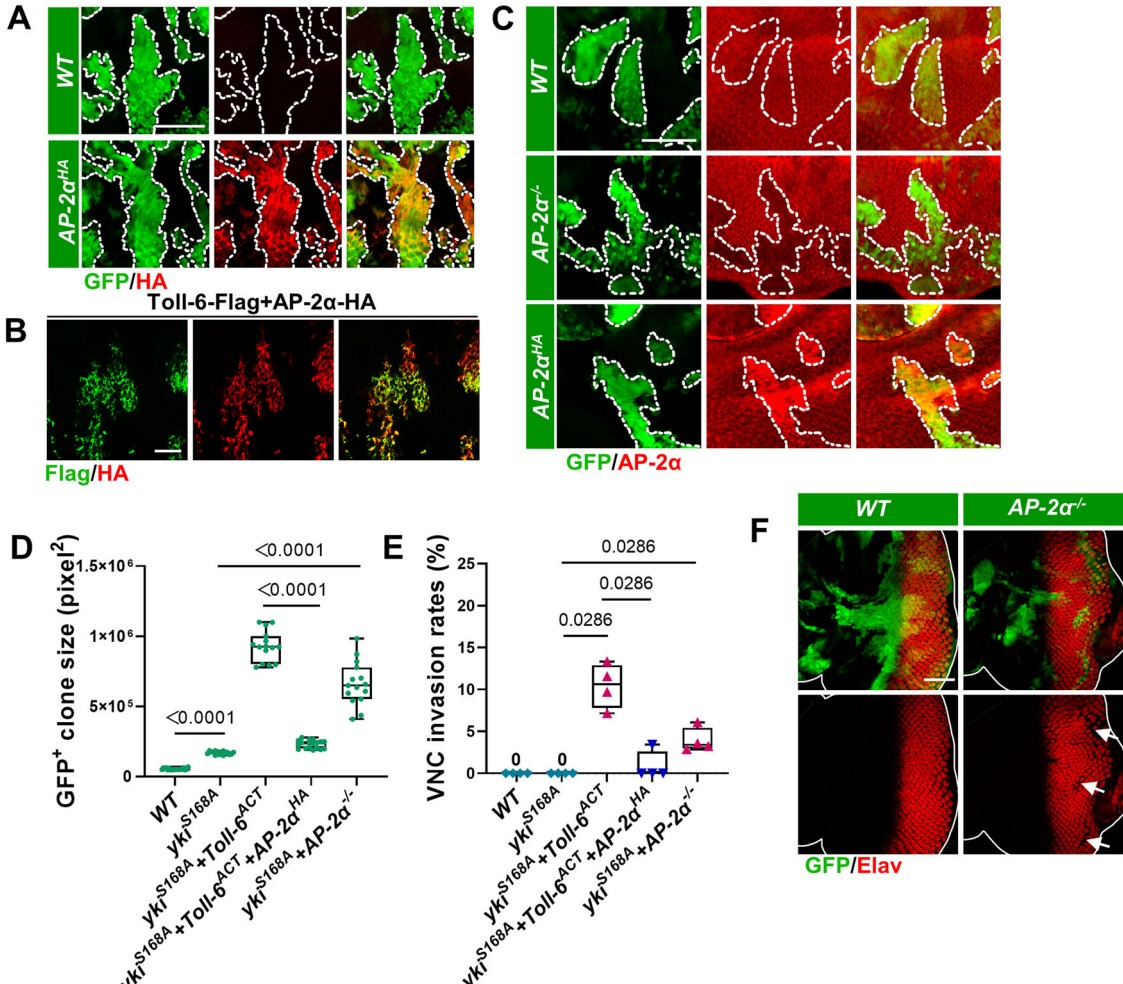

**Figure EV5. Toll-6 regulates Hedgehog signaling via AP-2α-dependent endocytosis defects.**

(A) E-A discs with GFP-labeled *WT* and *AP-2α^HA^* clones were stained with anti-HA antibody. (B) E-A discs with *Toll-6^ACT.Flag^/AP-2α^HA^* clones, co-immunostained with anti-Flag and anti-HA antibodies. (C) E-A discs with GFP-labeled *WT*, *AP-2α^−/−^*, and *AP-2α^HA^* clones were stained with anti-AP-2α antibody. (D, E) Quantification of GFP-labeled clone size in E-A discs (D, from left to right, $n = 19, 16, 15, 18, 15$), or percentage of clone invasion into VNC (E, total number from four independent experiments, from left to right, $n = 92, 95, 115, 107, 127$) for the indicated genotypes, as related to Fig. 4H. (F) E-A discs with GFP-labeled *WT* (left) and *AP-2α^−/−^* (right) clones were stained with anti-Elav antibody, the arrows indicate decreased Elav protein level. The *P* values of (D, E) were determined by unpaired nonparametric Mann–Whitney test. Exact *P* values are shown in the figures. The box plots of (D, E) boundaries represent the 25th (lower quartile) and 75th (upper quartile) percentiles, with the center line indicating the median, and the whiskers extend to the minimum and maximum values. Scale bars: 50 μm for (A, C), and (F); 20 μm for (B).

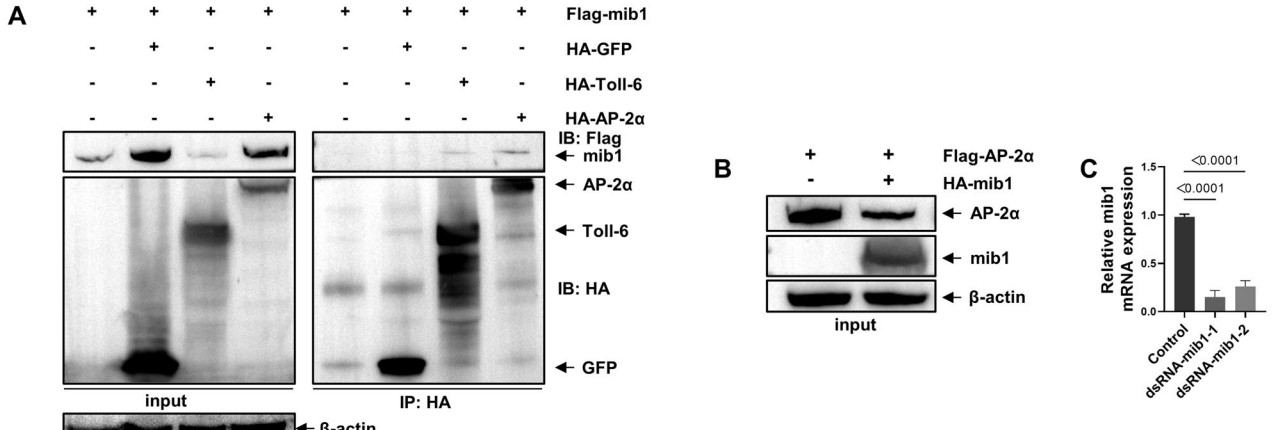

**A**

Flag-mib1
HA-GFP
HA-Toll-6
HA-AP-2α

IB: Flag — ← mib1
← AP-2α
← Toll-6
IB: HA
← GFP

input    IP: HA

← β-actin

**B**

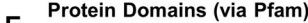
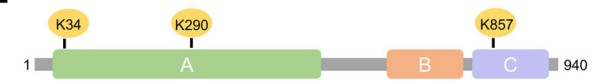

Flag-AP-2α
HA-mib1

← AP-2α
← mib1
← β-actin

input

**C**

Relative mib1 mRNA expression

<0.0001
<0.0001

Control   dsRNA-mib1-1   dsRNA-mib1-2

**D**  MusiteDeep results for AP-2α

| ID | Position | Residue | PTMscores | Cutoff=0.45 |
|---|---|---|---|---|
| sp\|P91926\|AP2A_DROME | 34 | K | Ubiquitination:0.48 | 0.48 |
| sp\|P91926\|AP2A_DROME | 290 | K | Ubiquitination:0.502 | 0.502 |
| sp\|P91926\|AP2A_DROME | 857 | K | Ubiquitination:0.453 | 0.453 |

**E**  Protein Domains (via Pfam)

K34    K290    K857

1    A    B    C    940

**A (29-587): Clathrin/coatomer adapt-like_N**
**B (714-818): Clathrin_a/b/g-adaptin_app_Ig**
**C (827-934): Clathrin_a-adaptin_app_sub_C**

**F**

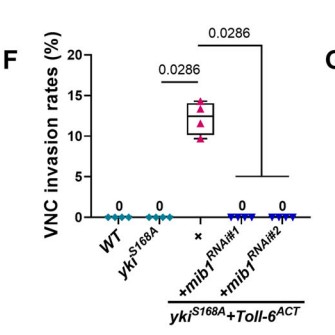

VNC invasion rates (%)

0.0286
0.0286

0   0   +   0   0

WT   yki^S168A   +mib1^RNAi#1   +mib1^RNAi#2

yki^S168A+Toll-6^ACT

**G**

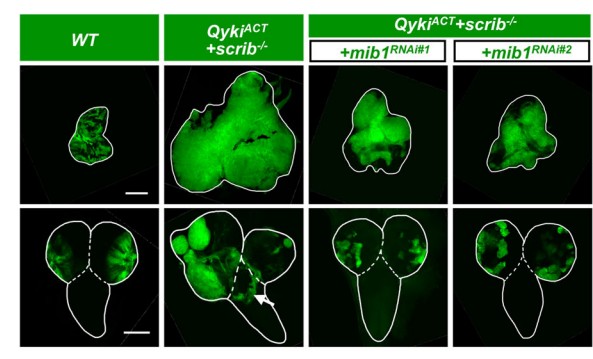

WT    Qyki^ACT+scrib^-/-    Qyki^ACT+scrib^-/-
                            +mib1^RNAi#1    +mib1^RNAi#2

**H**

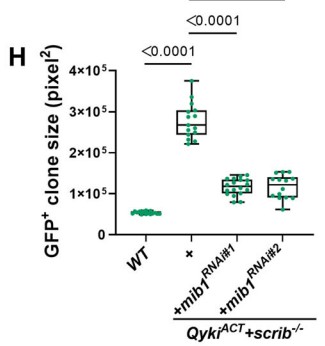

GFP+ clone size (pixel²)

<0.0001
<0.0001
<0.0001

WT   +   +mib1^RNAi#1   +mib1^RNAi#2

Qyki^ACT+scrib^-/-

**I**

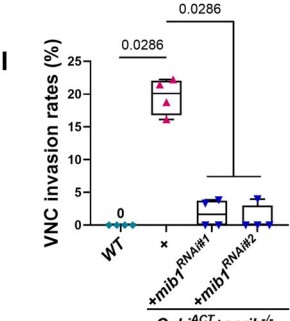

VNC invasion rates (%)

0.0286
0.0286

0   +   +mib1^RNAi#1   +mib1^RNAi#2

WT

Qyki^ACT+scrib^-/-

◀  **Figure EV6.   Toll-6 physically interacts with Mib1 to degrade AP-2α.**

(A) S2 cells transfected with HA-tagged AP-2α, HA-tagged Toll-6 and Flag-tagged Mib1. Mib1 physically interacts with Toll-6 and AP-2α. (B) The lysates were analyzed by western blotting. Mib1 can reduce the protein expression of AP-2α. (C) qRT-PCR analysis showing efficiency of dsRNA-mediated depletion of Mib1. (D) The detailed information on ubiquitination sites identified through MusiteDeep. Three AP-2α sites (K34, K290, and K857) were identified in the framework (Cut off = 0.45). (E) Distribution of AP-2α sites identified on Pfam. K34 and K290 are located in the N-terminal region, whereas K857 is located in C-terminal region. (F) Quantification of the percentage of clone invasion into VNC (total number from four independent experiments, from left to right, $n = 74, 95, 115, 142, 113$) for the indicated genotypes. (G) E-A discs (top) and VNC (bottom) with ey-FLP/QMARCM-induced GFP-labeled clones of *WT*, *Qyki^{ACT}/scrib^{-/-}*, *Qyki^{ACT}/scrib^{-/-}/mib1^{RNAi#1}*, and *Qyki^{ACT}/scrib^{-/-}/mib1^{RNAi#2}*, the arrow indicates VNC invasion site. (H, I) Quantification of GFP-labeled clone size in E-A discs (H, from left to right, $n = 16, 15, 18, 14$), or percentage of clone invasion into VNC (I, total number from four independent experiments, from left to right, $n = 88, 118, 109, 102$) for the indicated genotypes. The P value of (C) was determined using one-way ANOVA with Tukey's multiple comparison test; the P values of (F, H, I) were determined by unpaired nonparametric Mann–Whitney test. Exact P values are shown in the figures. The box plots of (F, H, I) boundaries represent the 25th (lower quartile) and 75th (upper quartile) percentiles, with the center line indicating the median, and the whiskers extend to the minimum and maximum values. Scale bars: 200 μm for (G).

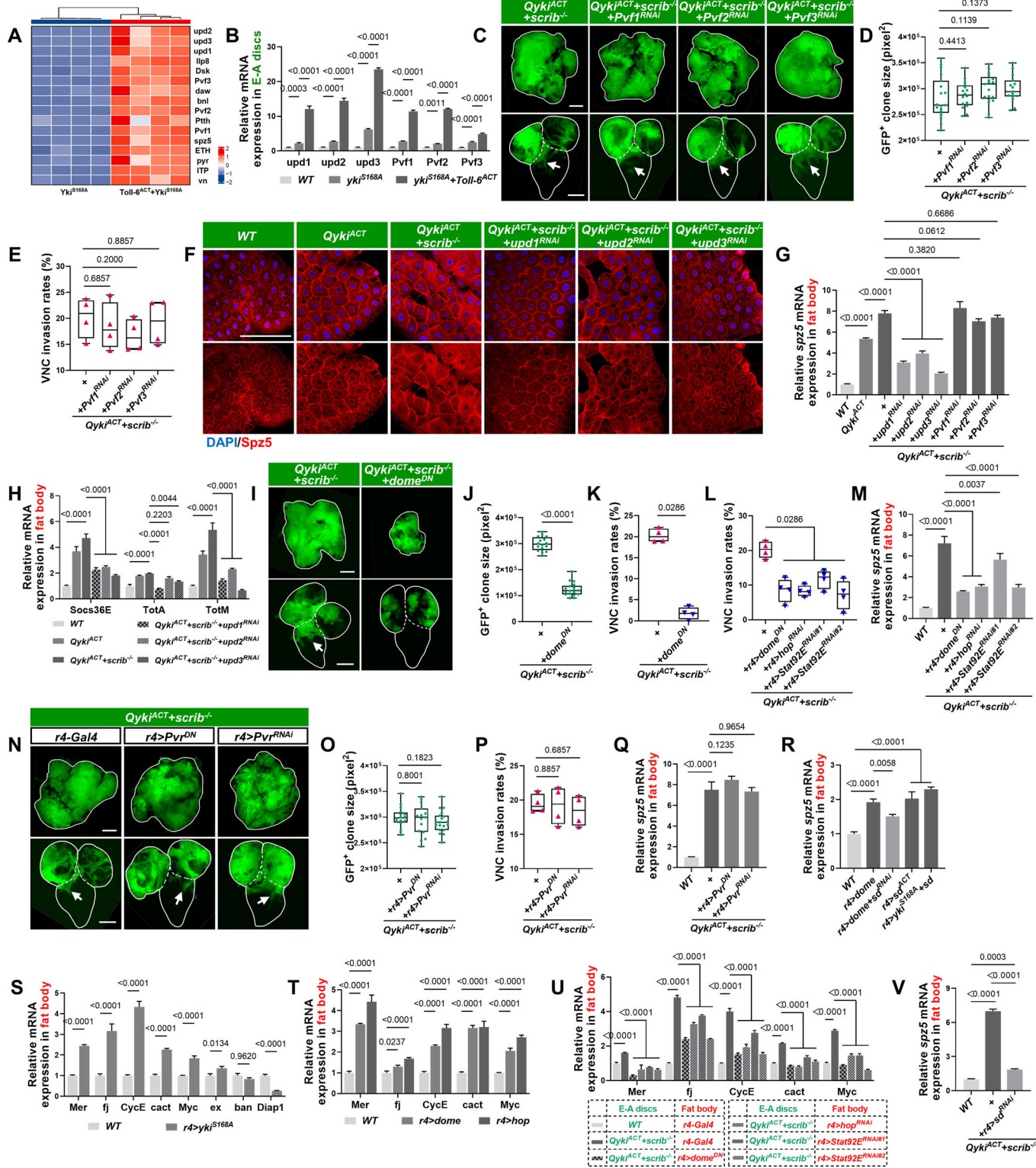

◀ **Figure EV7. Tumor-derived Upds promote Spz5 secretion from the fat body.**

(A) Heatmap of selected top differentially expressed *Drosophila* secreted ligands between the *yki*^S168A^ and *yki*^S168A^/*Toll-6*^ACT^ samples. Normalized gene expression values were calculated as log2(CPM + 1) and scaled by column. (B) Quantitative real-time PCR (qRT-PCR) to determine the mRNA levels of indicated genes in E-A discs with *WT*, *yki*^S168A^, or *yki*^S168A^/*Toll-6*^ACT^ genotypes (*n* = 3 independent experiments). (C) E-A discs (top) and VNC (bottom) with ey-FLP/QMARCM-induced GFP-labeled clones of *Qyki*^ACT^/*scrib*^−/−^, *Qyki*^ACT^/*scrib*^−/−^/*Pvf1*^RNAi^, *Qyki*^ACT^/*scrib*^−/−^/*Pvf2*^RNAi^, and *Qyki*^ACT^/*scrib*^−/−^/*Pvf3*^RNAi^. Arrows indicate VNC invasion sites. (D, E) Quantification of GFP-labeled clone size in E-A discs (D, from left to right, *n* = 19, 16, 17, 15), or percentage of clone invasion into VNC (E, total number from four independent experiments, from left to right, *n* = 230, 214, 229, 236) for the indicated genotypes. (F) Fat bodies dissected from larvae bearing ey-FLP/QMARCM clones were stained with anti-Spz5 antibody. (G) qRT-PCR to determine spz5 mRNA levels in fat bodies with the indicated genotypes (*n* = 3 independent experiments). (H) qRT-PCR to determine the mRNA levels of Socs36E, TotA, and TotM in fat bodies with the indicated genotypes (*n* = 3 independent experiments). (I) E-A discs (top) and VNC (bottom) with ey-FLP/QMARCM-induced GFP-labeled clones of *Qyki*^ACT^/*scrib*^−/−^ and *Qyki*^ACT^/*scrib*^−/−^/*dome*^DN^, the arrow indicates VNC invasion site. (J, K) Quantification of GFP-labeled clone size in E-A discs (J, from left to right, *n* = 16, 18), or percentage of clone invasion into VNC (K, total number from four independent experiments, from left to right, *n* = 227, 240) for the indicated genotypes. (L) Quantification of the percentage of clone invasion into VNC with indicated genotypes (total number from four independent experiments, from left to right, *n* = 234, 199, 242, 217, 196), related to Fig. 6I. (M) qRT-PCR to determine spz5 mRNA levels in fat bodies with the indicated genotypes (*n* = 3 independent experiments), related to Fig. 6I. (N) E-A discs (top) and VNC (bottom) with GFP-labeled *Qyki*^ACT^/*scrib*^−/−^ clones induced by ey-FLP/QMARCM, and expression of *WT*, *Pvr*^DN^, and *Pvr*^RNAi^ in fat bodies under the control of *r4-Gal4*; the arrows indicate VNC invasion sites. (O, P) Quantification of GFP-labeled clone size in E-A discs (O, from left to right, *n* = 16, 15, 19), or percentage of clone invasion into VNC (P, total number from four independent experiments, from left to right, *n* = 220, 220, 222) for the indicated genotypes. (Q) qRT-PCR to determine spz5 mRNA levels in fat bodies with the indicated genotypes (*n* = 3 independent experiments). (R) qRT-PCR to determine the spz5 mRNA levels in fat bodies under the control of an *r4-Gal4* promoter to express *WT*, *dome*, *dome/sd*^RNAi^, *sd*^ACT^, and *yki*^S168A^/*sd* (*n* = 3 independent experiments). (S) qRT-PCR to determine the mRNA levels of Hippo pathway target genes in fat bodies under the control of an *r4-Gal4* promoter to express *WT* or *yki*^S168A^ (*n* = 3 independent experiments). (T) qRT-PCR to determine the mRNA levels of Hippo pathway target genes in fat bodies under the control of an *r4-Gal4* promoter to express *WT*, *dome*, or *hop* (*n* = 3 independent experiments). (U) qRT-PCR to determine the mRNA levels of Hippo pathway target genes in fat bodies with the indicated genotypes (*n* = 3 independent experiments), with E-A disc genotypes labeled in green and fat body genotypes labeled in red. (V) qRT-PCR to determine spz5 mRNA levels in fat bodies with the indicated genotypes (*n* = 3 independent experiments), related to Fig. 6O. The *P* values of (B, G, H, M, Q, R, S, T, U, V) were determined using one-way ANOVA with Tukey's multiple comparison test; the *P* values of (D, E, J, K, L, O, P) were determined by unpaired nonparametric Mann–Whitney test. Exact *P* values are shown in the figures. The box plots of (D, E, J, K, L, O, P) boundaries represent the 25th (lower quartile) and 75th (upper quartile) percentiles, with the center line indicating the median, and the whiskers extend to the minimum and maximum values. Scale bars: 200 μm for (C, F, I, N). DAPI 4′,6-diamidino-2-phenylindole.

