## [Peer Review File · The EMBO Journal]

Adipose tissue-secreted Spz5 promotes distal tumor progression via Toll-6-mediated Hh pathway activation in *Drosophila*

Xianjue Ma, Du Kong, Xiaoqin Li, Sihua Zhao, Chenliang Wang, Zixin Cai, Sha Song, Yifan Guo, Xiaoyu Kuang, Xianping Wang, Wenhan Liu, Peng Liu, Xiaowei Guo, Wenyan Xu, Yirong Wang, Bin Zhao, Bin Jin, and Li He

Corresponding authors: Xianjue Ma (maxianjue@westlake.edu.cn), Li He (lihe19@ustc.edu.cn), Bin Jin (jinbin@sdu.edu.cn), Du Kong (kongdu@email.sdu.edu.cn)

Review Timeline:

Submission Date:	1st Apr 25
Editorial Decision:	13th May 25
Revision Received:	19th May 25
Accepted:	4th Jun 25

Editor: Ieva Gailite

Transaction Report:

Please note that the initial review process for this manuscript took place with another journal. The initial reviewers' comments and authors' responses for this article have been made available.

Response to reviewers' comments:

Dear reviewers and editor,

Thank you very much for your time and insightful comments regarding our manuscript. We have conducted extensive additional experiments and incorporated the necessary revisions to address the concerns raised. The following is a point-by-point response to the reviewers' comments. For your convenience, we have highlighted the changes made in the revised manuscript using red font. We sincerely hope that you will find the revised manuscript satisfactory.

Response to reviewer's comments:**Reviewer #1:**

In this manuscript, Kong et al utilize Drosophila larval models of intestinal tumorigenesis to further elucidate fat body derived Spz5 on epithelial tumour growth and invasion. Using a dual genetic model system for independent genetic manipulation of tumour and its microenvironment (fat body and hemocytes) they find that the combination of loss of the polarity regulator Scribble (scrib^{-/-}) and gain of function of Yki/Yap (yki^{act}) leads to invasive tumour with upregulation of Toll-6 and Hh pathway activation. Through biochemical and genetic experiments, they authors identify a mechanism linking these various observations: Toll-6 regulates Hh signaling activity via association to a component of the ubiquitin degradation machinery (Mib-1) and ubiquitin-dependent degradation of the endosomal adaptor protein AP2.

This is a well written and executed manuscript containing a concise and complete data set and which expands from previous work by the authors on the role of fat body Spz5 and Toll in the elimination of scrib^{-/-} tumours in imaginal discs and discovers a new mechanism mediating the regulation of signaling by the immune receptor Toll.

Response: We are grateful for your recognition of our work.

Major comments:

1- The overall level of conceptual novelty of the study is questionable and there is an alarming lack of acknowledgement by the authors of pioneering work directly related to the central subjects of this paper:

a- The concept of the fat body as an epithelial tumour regulatory organ dates well back to the citations indicated by the authors

<https://www.sciencedirect.com/science/article/pii/S2211124714000734>

b- The type of genetic model system used here is not novel and has been reported before for the study of interorgan communication work in Drosophila.

<https://pubmed.ncbi.nlm.nih.gov/34361081/>;

<https://www.sciencedirect.com/science/article/pii/S1534580721006389?via%3Dihub>

c- The concept of a tumour promoter versus tumour suppressor role of host immunity influenced by the genetic characteristics of a tumour has been documented in work also omitted in the citations <https://pubmed.ncbi.nlm.nih.gov/20627081/>

Response: Thank you for your valuable feedback and for bringing these important references to our attention. We acknowledge that the concept of the fat body as an epithelial tumour regulatory organ has been previously established, as indicated by the reference provided (Reference *a*). We included this citation in our manuscript and explicitly recognize the pioneering work in this area. We also agree that the genetic model system used in our study is not entirely novel and has been utilized in previous research on interorgan communication in *Drosophila*, however, the molecular mechanisms we identified, both within the tumor and in the distal organs, are quite novel. We modified the text accordingly in our revised MS to

better contextualize our findings within the existing literature and to clearly articulate the novel contributions of our study. As for the Reference *c* you mentioned above, we have included this reference and discussed the relevance of these findings to our study, highlighting the novel aspects of our work that contribute to this ongoing research area.

Regarding the level of conceptual novelty of our study, we respectfully disagree with your assessment. We believe that our study is highly significant for the following reasons: **1)** Our study highlights the essential role of fat body-derived *spz5*, but not from macrophages, in regulating distal tumour progression, a finding that was previously unknown in the field. **2)** In addition to the previously published tumor suppressive role of Toll-6 in regulating tumor suppressive cell competition (Kong *et al.*, 2022, *Cell Rep*), our study reveals a remarkable tumour-promoting role of Toll-6 under a different genetic background. We also elucidate the molecular mechanism by which Toll-6 promotes cell-autonomous tumor progression, as well as systemic signal communication between the tumor and distal organs. **3)** Despite the documented crucial role of fat body in mediating the interorgan communication between distal tumors, our study reveals uncharacterized mechanisms involving multiple signaling pathways both in the tumors and the fat body (included in the revised Figs. 6 and S7). This not only uncovers a novel systematic understanding of tumour progression but also highlights the potential to target these organs for cancer treatment.

2- Toll and toll-like receptor signaling is well known to be controlled by endocytosis. Considering that, the authors should double check Toll-6 localization and downstream signaling in contexts where endocytosis is affected as it could have a direct impact on tumours via Toll-6 regulation rather than/in addition to Hh.

Response: We concur that TLR signaling is well established to be controlled by endocytosis. However, the data in this manuscript support a model wherein Toll-6 regulates endocytosis, rather than the other way around. Our genetic evidence clearly indicates that Toll-6 acts upstream of AP-2 α . **Firstly**, Toll-6^{ACT} clones exhibit defects in endocytosis, as shown in Fig. 4B. **Secondly**, overexpression of Toll-6^{ACT} markedly reduces the expression of AP-2 α (Fig. 4D), a central player in clathrin-mediated endocytosis, serving as a bridge between cargo recognition and vesicle formation. **Thirdly**, the tumorigenesis and Hh activation induced by *Toll-6^{ACT}+Yki^{ACT}* can be significantly suppressed by co-expressing AP-2 α (Figs. 4E-I). Therefore, our data suggest that Toll-6 regulates endocytosis-mediated Hh activation through AP-2 α .

Additionally, as shown below, our biochemical data demonstrate that Toll-6 physically interacts with AP-2 α (Figs. 4C, 5A), regulating AP-2 α stability and subsequent Hh signaling activation in a ubiquitin-dependent manner involving E3 ubiquitin ligase Mib1 (Fig. 5A-F).

We concur with your commentary that it is essential to examine whether Toll-6 localization is altered when endocytosis is impaired. Given the low endogenous expression level of Toll-6 (A), we analyzed the expression of exogenously expressed Toll-6 (wild-type) in clones where endocytosis is disrupted by mutating *Avl*, *Vps36*, or *TSG101*, which are essential components of the endocytic pathway and vesicle trafficking. Notably, in clones

expressing *Toll-6^{WT}*, Toll-6 predominantly localized to the plasma membrane (**B**). Conversely, in clones with disrupted endocytosis, Toll-6 staining was mainly observed in the cytosomes (**C-E**), consistent with the notion that TLR signaling can be controlled by endocytosis. Collectively, we appreciate your comments and these data indeed reveal a complex interplay between Toll-6 and endocytosis; additional experiments are necessary to further elucidate this relationship, although it is not the major focus of the current manuscript.

3- Related to the above, why the focus on Hh signaling? There are many other well characterized conserved pathways which are highly oncogenic and also regulated by endocytosis, such as EGFR and Wnt, just to name a few. Have the authors tried these as mediators of tumorigenesis and invasion in their model?

Response: Thank you for raising this issue. As shown in the Figure below, we have previously performed small scale antibody screen to examine the activation of several of the growth regulating pathways within the *Toll-6^{ACT}* clones, including JNK, Hippo, Notch, Hh, and JAK-STAT pathways. Now we also examined additional growth-regulating pathways including EGFR and Wnt.

We have previously demonstrated that JAK-STAT pathway activation was not significantly affected upon the overexpression of *Toll-6^{ACT}*, whereas JNK signaling was moderately activated (**A-B**) (Kong *et al.*, 2022, *Cell Rep*). Furthermore, we also found that ectopic expression of *Toll-6^{ACT}* robustly downregulated multiple Yki targets, including *Diap1* (**C**) (Kong *et al.*, 2022, *Cell Rep*). Interestingly, we detected a significant increase in punctate-like accumulation of NICD (Notch Intracellular Domain) and NECD (Notch Extracellular Domain) within *Toll-6^{ACT}* clones (**D-E**). However, there were no significant changes observed in the expression of the Notch target gene *Cut* or its reporter NRE-GFP in *Toll-6^{ACT}* clones (**F-G**). We also examined EGFR activation using a pERK-specific antibody, as well as the expression of EGFR target genes, including *spi* and *pnt*. We observed a significant downregulation on the phosphorylated ERK in Toll-6 clones (**H**), while the target genes of EGFR remained unaffected (**I-K**). Conversely, *Toll-6^{ACT}* clones exhibited a marked upregulation of the key Hh pathway transcription factor *Ci*, as well as the transcriptional upregulation of several target genes of the Hh pathway, including *Ptc* and *dpp* (**L-N**), indicating the activation of the Hh pathway upon *Toll-6^{ACT}* overexpression.

(Kong et al., 2022, Cell Rep)

Following your suggestion, we also examined Wnt activation upon Toll-6 expression. Numerous studies have demonstrated that endocytosis plays an important regulatory role in Wnt signaling activation (reviewed in Brunt & Scholpp, *Cell Mol Life Sci.*, 2017). Notably, in contrast to the reduced Wnt activation, we observed that endocytosis-defective Toll-6^{ACT} clones exhibit significant upregulation of *Wg*, *Arm*, and *fz3* transcription (A-C). Consistent with the elevated activation of Wnt signaling, inhibiting the Wnt pathway by knocking down *arm* significantly blocked tumor overgrowth and invasive migration in both *Yki^{S168A}+Toll6^{ACT}* and *QYki^{ACT}+scrib^{-/-}* contexts (D-I). We agree that it would be equally interesting to further dissect the seemingly contradictory mechanisms by which Toll-6 regulates Wnt-dependent tumor progression; however, this will be beyond the scope of current manuscript. These data have been included in the revision (Fig. S5).

Reviewer #2:

The paper is subdivided into the following chapters/figures:

1. In the first figure, authors describe the experimental set-up and their success in producing scrib-/Yki-act tumors in the eye epithelium while manipulating gene activity in the fat body and hemocytes to unravel a role of fat body derived Spz5 ligand binding to Toll-6 receptor in malignant tumors to drive growth and invasiveness. Upregulation of Toll-6 in tumors in 1D appears to be non-autonomous to the clone, which should be discussed, and labels of Figure 1 (and others) are too small.

Response: We re-examined the staining pattern of Toll-6 in $QYki^{CA}+scrib^{-/}$ clones using confocal microscopy, as shown, we observed that Toll-6 expression levels are upregulated in a cell-autonomous manner.

2. In the second figure, authors present evidence that an activated form of Toll6 is able to synergize with Yorkie-activated to drive tumor growth and invasiveness. Typo in panel 2B (eg. invasion rates)

Response: We have corrected this typo in the revision.

3. In the third figure, authors unravel the impact of an activated form of Toll-6 on Hh signaling and the role of Hh signaling in growth and invasiveness. Genotypes (and not figure labels) should be included in 3E and H.

Response: Genotypes are included in the revised manuscript for Figure 3E and 3H.

4. In the fourth figure, experimental data on the role of Toll-6 in leading to impaired endocytosis by downregulating AP-2 α and, consequently, driving Smo membrane localization (and Hh signaling activation) is shown. Some typos in the text.

Response: We have corrected the typos in the revision.

5. Lastly, in Figure 5, authors identify the molecular mechanisms used by Toll6 to drive AP-2 α through the Mib1 ubiquitin ligase.

Besides the typos annotated in my list, I think this ms is well supported by the experimental data, is timely (identifies very nicely the molecular mechanisms used in the communication between Spz5 coming from the fat body to activate Toll 6 in the tumor) and exciting (interorgan communication in cancer).

Response: Thank you very much for recognizing the importance of our work.

I have only one question. Who drives expression of Spz5 in the fat body in the first place? In other terms, is it the tumor the one promoting the expression of Spz5 from the fat body? If so, which is the mechanism? This would nicely close the circle and give some context to the process of interorgan communication tumor-fat body-tumor. Otherwise, the ms just deals with the genetic and molecular mechanism by which Spz5 coming from the FB activates Toll6 in the tumor to drive tumorigenesis through Hh.

Response: Thank you for raising this very important issue. We fully agree that elucidating the underlying mechanisms is crucial for a better understanding of how systemic changes influence tumorigenesis.

We did extensive additional experiments to explore the molecular mechanisms by which *spz5* is transcriptional upregulated in fat bodies. Firstly, we performed bulk RNA-seq analyses to compare transcriptomic changes between *Yki^{S168A}* and *Yki^{S168A}+Toll-6^{ACT}* tumors. These analyses revealed a total of 16 significantly upregulated ligands in the *Yki^{S168A}+Toll-6^{ACT}* tumor, among which the Unpaired (Upd) family of cytokines, *upd1*, *upd2*, and *upd3* being the top three, showing at least a 34-fold increase.

Consistently, we detected dramatic transcriptional upregulation of *upd1/2/3* in both *QYki^{S168A}+scrib^{-/-}* and *Yki^{S168A}+Toll-6^{ACT}* tumors. Notably, we also observed transcriptional upregulation of *Pvf1/2/3* (Figure S7B), which are known to be secreted from tumors to induce cachexia-like symptoms in *Drosophila*.

Subsequently, we cell-autonomously inhibited these ligands and found that *upd1/2/3* inhibition significantly blocked the tumor overgrowth and invasion phenotype (A-C). In contrast, inhibiting *pfv1/2/3* had no significant effect (D-F).

We then evaluated whether tumor-secreted Upds (1/2/3) could be responsible for the increased expression of *spz5* in the distal fat bodies. Indeed, we observed that knockdown of *upd1/2/3* significantly impeded *QYki^{S168A}+scrib^{-/-}* tumor-induced upregulation of Spz5 in the fat bodies (see figure below). This suggests that tumor derived Upds are essential for the systemic induction of Spz5 in distal fat bodies.

The Upd family members are crucial ligands that activate the Janus Kinase/Signal Transducers and Activators of Transcription (JAK/STAT) signaling pathway in *Drosophila*. This conserved signaling pathway plays a pivotal role in various developmental processes and pathological conditions, such as immune responses, tissue homeostasis, and tumorigenesis. Upd ligands bind to their cognate receptors, Domeless (Dome), leading to the phosphorylation and activation of JAK. Activated JAK then phosphorylates STAT, which subsequently dimerizes and translocates to the nucleus to regulate target gene transcription. Notably, STAT activation is observed both cell-autonomously in tumors (A and B) and non-cell-autonomously in distal fat bodies (A and C). Furthermore, knockdown of *upds* within the tumors inhibited STAT signaling activation in the fat bodies, as demonstrated by the transcriptional downregulation of multiple STAT target genes (D). Additionally, consistent with the cell-autonomous increase in JAK-STAT activity, inhibiting the JAK-STAT pathway by expressing a dominant negative form of *dome* (*dome^{DN}*) within the tumors impeded tumor growth and invasion (E-G).

Next, we dissected the role of the JAK-STAT pathway in tumor-bearing fat bodies using dual expression systems. Notably, genetic inhibition of JAK-STAT pathway specifically in the fat bodies, by expressing *dome^{DN}*, or by knockdown of *hop* (JAK) or *Stat92E* (STAT), not only significantly impeded tumor-induced upregulation of Spz5 in these tissues (A-C) but also suppressed the overgrowth and invasion of *Qyki^{Sl68A}+scrib^{-/-}* tumors (A, D, E). Conversely, knockdown of *PDGF- and VEGF-receptor related (Pvr)*, the receptor tyrosine kinase of Pvf, within the fat bodies had no effect on tumor progression or *spz5* upregulation (F-I).

We then explored the mechanism by which *spz5* is transcriptionally upregulated in the tumor-bearing fat bodies. We failed to identify any potential binding sites of Stat92E on the 2kb promoter regions of *spz5*, suggesting that activation of JAK-STAT pathway may not directly regulate *spz5* transcription. JAK-STAT signaling activation has been shown to promote Hippo pathway-dependent hematopoietic cell proliferation (PMID: 28620086). Intriguingly, we identified multiple putative binding motifs of Scalloped (Sd), the downstream transcription factor of Hippo signaling. Our cleavage under targets and tagmentation (CUT&Tag) analysis in S2 cells confirmed that the Yki/Sd transcriptional complex directly binds to multiple sites within the *spz5* promoter region, indicating that Hippo pathway directly regulates *spz5* transcription.

In accordance with the genetic epistasis interaction between the JAK-STAT pathway and Hippo pathway, we observed that activation of JAK-STAT signaling under physiological condition in the fat bodies significantly upregulates Spz5 expression in a Sd-dependent manner (A-B). Conversely, ectopic expression of an activated form of Sd, or co-expression of Yki and Sd, is sufficient to upregulate Spz5 expression in the fat bodies (A-B). Additionally, several target genes of the Hippo pathway, including *Merlin* (*Mer*), *four-jointed* (*fj*), *Cyclin E* (*CycE*), *cactus* (*cact*), and *Myc* (C), are also upregulated upon genetic activation of JAK-STAT signaling in the fat bodies (D). More importantly, under tumor-induced pathological conditions, fat body-specific inhibition of the JAK-STAT signaling suppresses *QYki^{S168A}+scrib^{-/-}*-induced upregulation of Hippo target genes (E). Furthermore, the knockdown of *sd* specifically in the fat bodies inhibits *QYki^{S168A}+scrib^{-/-}*-induced tumor progression and Spz5 increase in the fat body (F-J).

In summary, our data suggest that tumor-derived Upds activate the JAK-STAT signaling pathway in the distal fat bodies, which subsequently inactivates the Hippo pathway, directly upregulating *spz5* transcription and thereby systemically promoting the progression of distal *QYki^{S168A}+scrib^{-/-}* tumors. These findings have been incorporated into the revision (Figs. 6 and S7).

Reviewer #3:

Main comments:

1. It is not clear how *spz5* becomes elevated in the fat body. Does the tumour cause that? Is it elevated in the circulation? The paper suggests that fat body is the main mediator, but what occurs in the fat body was not characterized. The mechanism by which *Spz5* gets activated in the fat body should be explored.

Response: Thank you for raising this important issue.

We did extensive additional experiments to explore the molecular mechanisms by which *spz5* is transcriptional upregulated in fat bodies. Firstly, we performed bulk RNA-seq analyses to compare transcriptomic changes between *Yki^{S168A}* and *Yki^{S168A}+Toll-6^{ACT}* tumors. These analyses revealed a total of 16 significantly upregulated ligands in the *Yki^{S168A}+Toll-6^{ACT}* tumor, among which the Unpaired (Upd) family of cytokines, *upd1*, *upd2*, and *upd3* being the top three, showing at least a 34-fold increase.

Consistently, we detected dramatic transcriptional upregulation of *upd1/2/3* in both *QYki^{S168A}+scrib-/-* and *Yki^{S168A}+Toll-6^{ACT}* tumors. Notably, we also observed transcriptional upregulation of *Pvf1/2/3* (Figure S7B), which are known to be secreted from tumors to induce cachexia-like symptoms in *Drosophila*.

Subsequently, we cell-autonomously inhibited these ligands and found that *upd1/2/3* inhibition significantly blocked the tumor overgrowth and invasion phenotype (A-C). In

contrast, inhibiting *pfv1/2/3* had no significant effect (D-F).

We then evaluated whether tumor-secreted Upds (1/2/3) could be responsible for the increased expression of *spz5* in the distal fat bodies. Indeed, we observed that knockdown of *upd1/2/3* significantly impeded *QYki^{S168A}+scrib^{-/-}* tumor-induced upregulation of Spz5 in the fat bodies (see figure below). This suggests that tumor derived Upds are essential for the systemic induction of Spz5 in distal fat bodies.

The Upd family members are crucial ligands that activate the Janus Kinase/Signal Transducers and Activators of Transcription (JAK/STAT) signaling pathway in *Drosophila*. This conserved signaling pathway plays a pivotal role in various developmental processes and pathological conditions, such as immune responses, tissue homeostasis, and tumorigenesis. Upd ligands bind to their cognate receptors, Domeless (Dome), leading to the phosphorylation and activation of JAK. Activated JAK then phosphorylates STAT, which subsequently dimerizes and translocates to the nucleus to regulate target gene transcription. Notably, STAT activation is observed both cell-autonomously in tumors (A and B) and non-cell-autonomously in distal fat bodies (A and C). Furthermore, knockdown of *upds* within the tumors inhibited STAT signaling activation in the fat bodies, as demonstrated by the transcriptional downregulation of multiple STAT target genes (D). Additionally, consistent with the cell-autonomous increase in JAK-STAT activity, inhibiting the JAK-STAT pathway by expressing a dominant negative form of *dome* (*dome^{DN}*) within the tumors impeded tumor growth and invasion (E-G).

Next, we dissected the role of the JAK-STAT pathway in tumor-bearing fat bodies using dual expression systems. Notably, genetic inhibition of JAK-STAT pathway specifically in the fat bodies, by expressing *dome^{DN}*, or by knockdown of *hop* (JAK) or *Stat92E* (STAT), not only significantly impeded tumor-induced upregulation of Spz5 in these tissues (A-C) but also suppressed the overgrowth and invasion of *Qyki^{Sl68A}+scrib^{-/-}* tumors (A, D, E). Conversely, knockdown of *PDGF- and VEGF-receptor related (Pvr)*, the receptor tyrosine kinase of Pvf, within the fat bodies had no effect on tumor progression or *spz5* upregulation (F-I).

We then explored the mechanism by which *spz5* is transcriptionally upregulated in the tumor-bearing fat bodies. We failed to identify any potential binding sites of Stat92E on the 2kb promoter regions of *spz5*, suggesting that activation of JAK-STAT pathway may not directly regulate *spz5* transcription. JAK-STAT signaling activation has been shown to promote Hippo pathway-dependent hematopoietic cell proliferation (PMID: 28620086). Intriguingly, we identified multiple putative binding motifs of Scalloped (Sd), the downstream transcription factor of Hippo signaling. Our cleavage under targets and tagmentation (CUT&Tag) analysis in S2 cells confirmed that the Yki/Sd transcriptional complex directly binds to multiple sites within the *spz5* promoter region, indicating that Hippo pathway directly regulates *spz5* transcription.

In accordance with the genetic epistasis interaction between the JAK-STAT pathway and Hippo pathway, we observed that activation of JAK-STAT signaling under physiological condition in the fat bodies significantly upregulates Spz5 expression in a Sd-dependent manner (A-B). Conversely, ectopic expression of an activated form of Sd, or co-expression of Yki and Sd, is sufficient to upregulate Spz5 expression in the fat bodies (A-B). Additionally, several target genes of the Hippo pathway, including *merlin* (*mer*), *four-jointed* (*fj*), *Cyclin E* (*CycE*), *cactus* (*cact*), and *myc* (C), are also upregulated upon genetic activation of JAK-STAT signaling in the fat bodies (D). More importantly, under tumor-induced pathological conditions, fat body-specific inhibition of the JAK-STAT signaling suppresses *QYki^{S168A}+scrib^{-/-}*-induced upregulation of Hippo target genes (E). Furthermore, the knockdown of *sd* specifically in the fat bodies inhibits *QYki^{S168A}+scrib^{-/-}*-induced tumor progression and Spz5 increase in the fat body (F-J).

In summary, our data suggest that tumor-derived Upds activate the JAK-STAT signaling pathway in the distal fat bodies, which subsequently inactivates the Hippo pathway, directly upregulating *spz5* transcription and thereby systemically promoting the progression of distal *QYki^{S168A}+scrib^{-/-}* tumors. These findings have been incorporated into the revision (Figs. 6 and

S7).

2. How does toll-6 activate MMP1 and metastasis? and if it is through Hh signalling, how does hh signalling cause metastasis.

Response: Thank you for raising this issue. We have previously demonstrated that Toll-6 promotes organotropic metastasis by activating JNK signaling, a key regulator of cell migration (Mishra-Gorur *et al.*, 2019, *Dis Model Mech*, PMID: 31477571). Given that *Mmp1* can be transcriptional activated by JNK (PMID: 17082773), we hypothesized that *Toll-6^{ACT}+Yki^{ACT}* tumors upregulate *Mmp1* and promote invasion in a JNK-signaling-dependent manner.

We further explored the genetic interactions between the JNK and Hh pathways in the *Toll-6^{ACT}+Yki^{ACT}* tumorigenic condition. We found that inhibiting JNK by expressing a dominant negative form of *bsk* (*bsk^{DN}*) did not affect *Toll-6^{ACT}+Yki^{ACT}*-induced *Ci* induction (A-B). Conversely, knockdown of *ci* significantly suppressed *Mmp1* upregulation (C-D), suggesting that *Ci* genetically acts upstream of JNK signaling. Consistent with this, overexpression of *ci* is sufficient to upregulate *Mmp1* expression (E-F), whereas inhibition of *ci* significantly suppressed *Toll-6^{ACT}*-induced JNK activation, including *Mmp1* and *TRE-DsRed* induction (E-H). Moreover, inhibition of JNK completely abolished the invasive migration to the ventral nerve cord induced by *Toll-6^{ACT}+Yki^{ACT}*, *QYki^{ACT}+scrib-/-*, and *Ci^{ACT}+Yki^{ACT}* (I-J). Collectively, these data demonstrate that the Hh pathway acts upstream of JNK to drive tumor cell invasion. These data have been incorporated into the revision (Fig. S4).

3. Is AP2 and endocytosis only regulating Hh signalling? It is unlikely to be the case. Besides, it has been shown previously that Yki activation in endocytosis-defective cells is accompanied by activation of the JNK signaling pathway to cause increased tumour malignancy (Moberg lab, 2011). This point is not so novel

Response: We acknowledge the reviewer's comment on the multifunctional nature of AP-2 α -mediated endocytosis, which indeed extends beyond the regulation of Hh signaling to pathways such as JNK signaling, as noted by Moberg's lab in 2011. It is important to clarify, however, that the central observation of Moberg's work was the JNK-dependent activation of Yki due to disruptions in the endolysosomal pathway, without extensive mechanistic detail. Our study differentiates from these findings by providing novel mechanistic insights. Specifically, we have demonstrated that clonal depletion of AP-2 α elevates Ci expression and potentially JNK activation (pending further testing), yet this alone does not precipitate tumorigenesis. Tumor formation was observed only with the concomitant expression of an activated form of Yki. In a similar vein, the malignant transformation induced by ectopically expressed *Toll-6^{ACT}* requires the presence of *Yki^{ACT}* for full manifestation. Neither the isolated activation of Hh signaling nor the sole perturbation of the endocytosis pathway was sufficient to initiate tumor formation in our models. Our genetic data, therefore, underscores the critical interplay between Hh signal activation and Yki hyperactivation in the promotion of tumorigenesis.

The focus of our research, as detailed in the manuscript, is to explore the specific nuances of how adipose-derived *Spz5* promotes tumor formation via Toll-6-mediated Hh activation and subsequent collaborative tumor-promoting interactions with *Yki^{ACT}*. Although the role of AP-2 α in endocytosis and signaling is established, the addition of extensive *in vivo* data elucidates novel mechanisms by which tumors systematically upregulate *spz5* transcription in fat bodies. Consequently, we believe that our current manuscript provides significant novel insights into the systemic interorgan communication affecting tumor progression and offers a new perspective on Hh and Hippo pathway-mediated tumor progression.

Minor comments:

Fig 2 D, MMP1 level should be quantified

Response: We have quantified the relative intensity of MMP1 in the revision.

Fig 2 G, cell death should be quantified.

Response: We have quantified the relative intensity of Dcp-1 in the revision (Fig. S2H).

Demarcation of VNC invasion is not clear; in 4J, VNC invasion should be quantified more objectively, and should be quantified.

Response: We have modified the related method section and re-quantified the VNC invasion rate in the revision.

Quantification of *in vivo* tumor invasion:

ey-FLP-induced MARCM or QMARCM clones not only appear in the eye-antennal imaginal discs but also expressed in the brain lobes under normal conditions. However, the adjacent ventral nerve cord (VNC) does not typically express these clones under normal conditions. Therefore, in *Drosophila* tumor experiments, the VNC is often examined to assess the tumor's invasive capability in the context of tumor induction in the eye-antennal imaginal discs. To achieve this, first use the MARCM or QMARCM technique to induce the formation of fluorescently labeled cell clones of various genotypes in the eye-antennal imaginal discs of *Drosophila* larvae. Second, dissect the VNC from these larvae and perform DAPI staining to distinguish between the brain lobes and the VNC. Conduct four independent replicate experiments, and for each experiment, calculate the ratio of VNCs that exhibit fluorescently labeled invasive clones, with at least 20 larvae per experiment (A).

Final comments from the editor after revision:

Here are some further comments to help you understand our reasoning. We noted previously that our decision was based on the limited data to support the mechanism of Spz5 activation in fat bodies and the evidence related to the role of Hh signaling pathway in endocytosis. Whilst we appreciate that you have added genetic inhibition of upd1/2/3 and JAK/STAT to show the effect on Spz5, we are concerned that this may not advance your conclusions on the mechanism to understand the regulation of Spz5 to the level to enable us to reconsider your paper. Therefore, at this point, it is not clear to me that the further work you wish to include will result in a successful further review of your work. Therefore, at this stage, we do not think the revisions will satisfy the reviewers.

Response: We have incorporated a substantial amount of additional data to elucidate the molecular mechanisms by which the fat body regulates spz5 transcription. However, for unknown reasons, the editor declined to send the manuscript back to the reviewers.

Dear Xianjue,

Thank you for submitting your manuscript together with the reviews from another journal and your point-by-point response to them to The EMBO Journal. I have now received input from an arbitrating advisor, who has evaluated the revised manuscript. I have copied the advisor's comments below. As you can see, he/she recommends publication of your revised study in our journal. Therefore, I will accept your manuscript after reformatting of the manuscript according to The EMBO Journal guidelines as listed below.

1. Please reduce the number of keywords to five.
2. Please check that the funding information is correct and identical both in the manuscript and our online system. Currently, HRHI program (1011103360222B1) of Westlake Laboratory of Life Sciences and Biomedicine, and "Pioneer" and "Leading Goose" R&D Program of Zhejiang (2024SSYS0034) are missing in our online system.
3. At EMBO Press we ask authors to provide source data for the main manuscript figures. You will receive a separate email with instructions for providing source data with your revised manuscript, including how to upload and organize the files.
4. Please upload the supplemental figures as individual figure files and rename them into Expanded View figures: figure EV1-EV7. Their legends should be added to the manuscript text, after the main figure legends.
5. CRedit has replaced the traditional author contributions section because it offers a systematic, machine-readable author contributions format that allows for more effective research assessment. Please remove the Authors Contributions from the manuscript and use the free text boxes beneath each contributing author's name in our online submission system to add specific details on the author's contribution. More information is available in our guide to authors.
6. Please rename "Declaration of interests" section into "Disclosure and competing interests statement" (further info: <https://www.embopress.org/page/journal/14602075/authorguide#conflictsofinterest>).
7. Please update references according to The EMBO Journal style - they should be listed in alphabetical order. Where there are more than 10 authors on a paper, the first 10 should be listed, followed by 'et al.' Please see further information here: <https://www.embopress.org/page/journal/14602075/authorguide#referencesformat>
8. All Materials and Methods need to be described in the main text using our 'Structured Methods' format. According to this format, the Methods section includes a Reagents and Tools Table (listing key reagents, experimental models, software and relevant equipment and including their sources and relevant identifiers) followed by a Methods and Protocols section describing the methods, ideally using a step-by-step protocol format. The aim is to facilitate adoption of the methodologies across labs. Please download and fill our Reagents and Tools Table template (.docx), which you can find in our author guidelines: <https://www.embopress.org/page/journal/14602075/authorguide#structuredmethods>
An example of a Method paper with Structured Methods can be found here: <https://www.embopress.org/doi/10.15252/msb.20178071>.
When submitting your revised manuscript, please upload it as a separate file choosing the file type "Reagent Table". The information currently provided in Key Resources Table could be adapted to this format.
9. Please rename Tables S1-S2 into Tables EV1-EV2 and update their callouts throughout the text. For Table EV1, please add the title and legend to the top of the page. For Table EV2, please add the title added to the legend.
10. Please rename Table S3/EV3 into Dataset EV1 and consider uploading it as an Excel file.
11. Figure 7 is not mentioned in the manuscript text; please add the corresponding callout.
12. Tables should be mentioned in the text in a numerical order. Currently, Table EV3 is mentioned before Table EV1, but this issue should resolve upon renaming of Table EV3 into Dataset EV1.
13. Please remove BioRender disclaimer from Acknowledgements and add to a dedicated section in the Methods section using the following format:
Graphics:
(some of the... OR Figure #... OR synopsis) Graphics were created with BioRender.com.
14. Please remove the list of supplemental information from the manuscript text file.
15. Our data editors have flagged the following issues in figure legends that need correcting:
 - Please provide the exact p values in the legends of figures 1B, C, E, G, H, N, O; 2B, C, E; 2B, C, E; 3E, 4F, 5H, 6A, C, D, E, G, H, J, K, L, N, P, Q, R .
 - 2. Please define the box plots in terms of center in the legends of figures 1B, C, E, G, H, N, O; 2B, C, E; 3E, H; 4F; 5H, 6C, D, E, G, H, J, K, N, P, Q, R.
 - 3. Please define the border in the legend of figures 1A, F, J, K, L, M; 2A, D, F, G; 3A, B, C, D, F, G; 4A, B, D, G, H; 5E, F, G; 6B, F, I, O.
 - Please note define the white arrows in the legend of figures 1A, 3A, C.
16. Papers published in The EMBO Journal are accompanied online by a 'Synopsis' to enhance discoverability of the manuscript. Please submit a short (1-2 sentences) summary of the findings and their significance in addition to the already provided bullet points highlighting the key results. Please also send us a synopsis image that is 550x300-600 pixels large (width x height, jpeg or png format). You can either show a model or key data in the synopsis image. Please note that the image size is rather small and that text needs to be readable at the final size.

Please let me know if you have any questions regarding any of these points. You can use the link below to upload the revised

files.

With best wishes,

Ieva

We realize that it is difficult to revise to a specific deadline. In the interest of protecting the conceptual advance provided by the work, we recommend a revision within 3 months (11th Aug 2025). Please discuss the revision progress ahead of this time with the editor if you require more time to complete the revisions.

Referee #1:

Authors have satisfactorily addressed all the concerns raised by the three main reviewers (specially the Upd/JAK-STAT connection) and as such, the ms is ready for publication.

The authors addressed the remaining editorial issues.

Dear Xianjue,

Thank you for addressing the final editorial requests and for approving the proposed textual edits. I am now pleased to inform you that your manuscript has been accepted for publication.

Finally, we would like to promote your manuscript among the Chinese readership. Therefore, we would like to invite you to prepare a short summary of the manuscript in Chinese (1500-2000 Chinese characters), which we will promote on the WeChat platform 'BioArt' with more than 610,000 followers.

If you are interested in this opportunity, we recommend covering the article very close to its online publication date. Thus, ideally we would very much appreciate if you could send us a draft within the next 7 working days. Please let us know whether or not you would be interested in contributing such a short summary in Chinese.

I have included below some general guidelines on how to prepare a summary and a link to recent examples for your reference. Please let me know if you have any questions about this.

If you have any questions, please do not hesitate to contact the Editorial Office. Thank you for this contribution to The EMBO Journal and congratulations on a nice study!

With best wishes,

Ieva

General WeChat Summary Guidelines

1. These summary articles are meant to be targeting general audience so please limit the use of specialized technical terms, acronyms and jargon.
2. A summary usually starts with brief background information of the reported work, which is followed by explaining the findings in some detail, and ends with a short review of the conclusions as well as the implications of the work and future directions for the research.
3. The summary should contain a visual abstract, which can be the one provided in the paper.
4. Please provide ONE SINGLE document containing all text and graphical materials, ideally as a Word .docx or .doc file. Please DO NOT provide the document as a .pdf file.
5. Please DO NOT publicly release the document before the paper is officially published online.

Summary Examples

EMBO Journal | 灵珠与魔丸：昆虫miRNA调控病毒感染虫媒和植物的双重作用

EMBO Mol Med | 陈良/舒红兵合作揭示毛花甘C通过STUB1-FOXP3促进抗肿瘤免疫的机制

EMBO Rep | 王一飞/郑楷/刘凯胜团队揭示HDAC6调控cGAS-STING介导的抗病毒免疫的分子机制
